# AlgoTune: Can Language Models Speed Up General-Purpose Numerical Programs?

**Ori Press**[1]   **Brandon Amos**[3]   **Haoyu Zhao**[2]   **Yikai Wu**[2]   **Samuel K. Ainsworth**
**Dominik Krupke**[4]   **Patrick Kidger**[5]   **Touqir Sajed**[6]   **Bartolomeo Stellato**[2]
**Jisun Park**[2,7]   **Nathanael Bosch**[1]   **Eli Meril**[8]   **Albert Steppi**[9]
**Arman Zharmagambetov**[3]   **Fangzhao Zhang**[10]   **David Pérez–Piñeiro**[11]   **Alberto Mercurio**[12]
**Ni Zhan**[2]   **Talor Abramovich**[8]   **Kilian Lieret**[2]   **Hanlin Zhang**[13]
**Shirley Huang**[13]   **Matthias Bethge**[1]   **Ofir Press**[2]

[1] Tübingen AI Center, University of Tübingen   [2] Princeton University   [3] Meta (FAIR)
[4] TU Braunschweig   [5] Cradle Bio   [6] LG Electronics Canada
[7] Seoul National University   [8] Tel Aviv University   [9] Quansight PBC
[10] Stanford University   [11] Norwegian University of Science and Technology   [12] EPFL   [13] Harvard University

## Abstract

Despite progress in language model (LM) capabilities, evaluations have thus far focused on models' performance on tasks that humans have previously solved, including in programming (Jimenez et al., 2024) and mathematics (Glazer et al., 2024). We therefore propose testing models' ability to design and implement algorithms in an open-ended benchmark: We task LMs with writing code that efficiently solves computationally challenging problems in computer science, physics, and mathematics. Our AlgoTune benchmark consists of 154 coding tasks collected from domain experts and a framework for validating and timing LM-synthesized solution code, which is compared to reference implementations from popular open-source packages. In addition, we develop a baseline LM agent, AlgoTuner, and evaluate its performance across a suite of frontier models. AlgoTuner uses a simple, budgeted loop that edits code, compiles and runs it, profiles performance, verifies correctness on tests, and selects the fastest valid version. AlgoTuner achieves an average $1.72\times$ speedup against our reference solvers, which use libraries such as SciPy, sk-learn and CVXPY. However, we find that current models fail to discover algorithmic innovations, instead preferring surface-level optimizations. We hope that AlgoTune catalyzes the development of LM agents exhibiting creative problem solving beyond state-of-the-art human performance.

## 1   Introduction

Language models have become increasingly capable at tasks in programming and mathematics (Liu et al., 2024; Anthropic, 2025a; Google DeepMind, 2025). But the research community has mostly focused on studying their ability to write simple standalone functions from scratch, as in HumanEval (Chen et al., 2021), MBPP (Austin et al., 2021), LiveCodeBench (Jain et al., 2024), and Aider Polyglot (Gauthier, 2024); or to fix bugs in existing software libraries, as in SWE-bench (Jimenez et al., 2024).

These benchmarks challenge AI systems with problems that have previously been solved by humans. For example, SWE-bench tasks consist of fixing historical bugs on GitHub, all of which have been fixed. HumanEval and the similar followups task LMs with reproducing human-written code based on given descriptions.

39th Conference on Neural Information Processing Systems (NeurIPS 2025) Track on Datasets and Benchmarks.

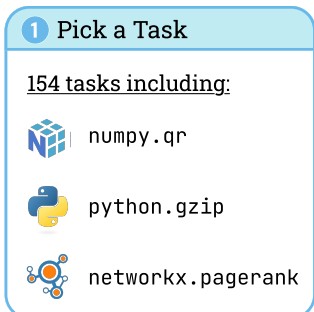
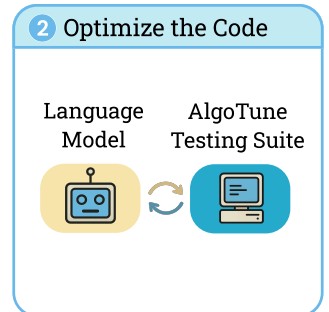
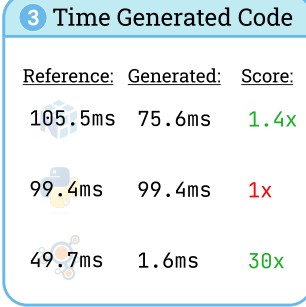

Figure 1: AlgoTune challenges LMs to optimize 154 numerical functions, including QR Decomposition, gzip Compression and PageRank. We score LMs based on how much faster their generated code is than reference solvers. For concrete examples, see §4.2.

But what if AI could go beyond what has already been done? What if AI systems could take optimized code from a popular Python library like Numpy (Harris et al., 2020), SciPy (Virtanen et al., 2020) or NetworkX (Hagberg et al., 2008) and make it ever faster?

To measure this, we introduce AlgoTune, a benchmark designed to assess the ability of LM systems to optimize the runtime of 154 functions in a wide variety of domains including Math (Cholesky factorization, Matrix exponential), Science (The FitzHugh–Nagumo ordinary differential equations, Heat Equation), Computer Science (SHA256 hashing, Graph Isomorphism, KD Tree Construction, gzip compression), and Machine Learning (Lasso regression, Kalman filters).

Existing code benchmarks contain tasks that have a binary outcome: the model either solves the task at hand – programming a specified function or fixing a bug – or it fails to do so. In AlgoTune, we score systems based on the speed of their synthesized code relative to a reference implementation, a metric with no absolute upper bound. We argue that AlgoTune narrows the gap between benchmark objectives and real-world goals: success on this benchmark could translate to tangible performance gains in widely used numerical libraries.

To improve the speed of a reference function, language models can utilize a variety of techniques, including implementing faster algorithms or rewriting the code in a lower-level language like C. Finding a faster algorithm may in some cases involve searching in the existing literature: one of our tasks uses SciPy's `spatial.kd_tree`, which uses the algorithm from Maneewongvatana and Mount (1999), but more recent works such as Muja and Lowe (2014) have proposed faster algorithms. In other cases, optimizing the given reference algorithm might require the LM to discover a novel approach.

In addition to our benchmark, we propose an LM agent, AlgoTuner, that iteratively develops efficient code for AlgoTune tasks. To enable writing efficient code, we equip AlgoTuner with tools including Cython (Behnel et al., 2011) and Numba (Lam et al., 2015) (see Appendix D for the complete list). When running on o4-mini-high, AlgoTuner is able to optimize code for 59.7% of the AlgoTune tasks, yielding an AlgoTune score of 1.72×. As we show in §4, these speedups are minor surface-level optimizations. We did not observe AlgoTuner finding any novel algorithmic improvements across any of the LMs tested.

Our contributions are threefold: 1) a benchmark that challenges LMs to optimize 154 functions in a wide variety of domains, including functions from popular open-source repositories, 2) a test suite that allows for robustly testing and timing AI-synthesized code for correctness, and 3) the AlgoTuner agent that can iteratively attempt to optimize a given function using any frontier LM.

Concurrently, KernelBench (Ouyang et al., 2025) challenges LMs to develop CUDA GPU kernels, evaluated on their speed. They focus on narrow, specific PyTorch operations such as `Matmul_Add_Swish_Tanh_GELU_Hardtanh` or neural network blocks such as `EfficientNetB2`, whereas AlgoTune contains a range of numerical functions across a wide array of domains.

We hypothesize that further work in this direction may lead to a future in which LMs are used to autonomously write highly-optimized code, potentially via novel algorithmic discovery. Our software

makes this process simple: users enter a function of interest, write an input data generator and output verifier, and get back an optimized version of their code.

## 2 The AlgoTune Benchmark

This section defines the benchmark scope (domains, taxonomy, and design principles) (§2.1), explains task construction (generators, reference solvers, verifiers, and QA) (§2.2), and details the evaluation protocol (instance sizing, timing, metrics, splits, and budget) (§2.3). Together, these subsections specify how tasks are generated, how correctness and speedups are measured, and how results can be reproduced.

### 2.1 Benchmark Scope

The AlgoTune benchmark consists of 154 tasks, challenging AI systems to write performant code for a variety of numerical and scientific problems, such as Graph Coloring, Spectral Clustering, and the Wasserstein distance function. We score an AI system based on how much faster its generated implementations are relative to our reference implementations. Our reference implementations use functions from popular libraries such as NumPy (Harris et al., 2020), SciPy (Virtanen et al., 2020), and NetworkX (Hagberg et al., 2008).

AlgoTune's test suite includes a *solution verifier* that runs an AI-generated implementation on a held-out test set of inputs to ensure correctness, and a *runtime profiler* that measures the wall-clock execution time of the code. The AI system's score for each task is the multiple of how much faster the synthesized code is than the reference implementation on the test input set.

Where possible, our reference implementations are simply calls to functions from popular Python packages because 1) these repositories contain extensive test suites, increasing confidence in the reference implementation correctness and 2) their popularity makes it so that further optimizations to these functions may have a direct real-world impact.

**Data Contamination.** Most existing benchmarks consist of a test set of questions and answers, and if this set leaks into training data, it causes the resulting LM to perform well on the test set without extrapolating to novel, real-world queries (Dong et al., 2024). AlgoTune bypasses this issue by not having *answers*. Instead we rely on reference solvers that are already publicly known and which we also include in the prompt shown to agents solving AlgoTune tasks.

**Can functions in popular Python repositories be sped up?** NumPy (Harris et al., 2020), SciPy (Virtanen et al., 2020) and NetworkX (Hagberg et al., 2008) are three popular Python libraries we use for many reference solvers in AlgoTune. They contain highly efficient code that has been developed by thousands of contributors. Given this, it may seem unlikely that significant performance improvements are possible in these libraries. However, Appendix E presents a sample of merged pull requests from the past two years in the above repositories. The runtime benchmark results presented in those pull requests show performance improvements that range from 2.7x speedups, to over 600x speedups. Even in highly optimized libraries substantial performance headroom remains.

**Benchmark construction.** To build our benchmark, we recruited 21 contributors, both committers to the software packages used in AlgoTune, as well as academics who contributed tasks from their field of expertise. Contributors were asked to submit reference solvers, input data generators and solution verifiers. Each submitted task was then reviewed by two other contributors. The AlgoTune test set consists of 154 tasks in 13 categories (see Appendix A for the full list of tasks). This is comparable to previous benchmarks including HumanEval (Chen et al., 2021), Bamboogle (Press et al., 2022), and Plot2Code (Wu et al., 2024), which contain 164, 125, and 132 tasks respectively. Unlike previous benchmarks, where every task instance can either be solved or not, since every task in AlgoTune can always be further optimized, we believe that our 154 tasks provide a suitable environment to benchmark LMs. In addition, we provide a development set intended for use in agent prototyping. Our development set consists of 5 tasks.

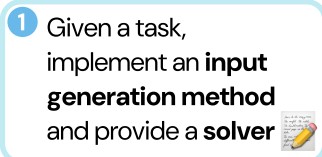 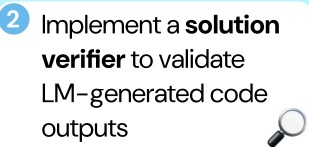 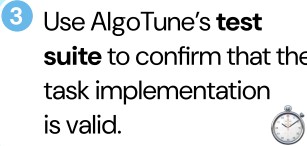

Figure 2: The task collection pipeline for AlgoTune. We define an input generation method and solver for each task, along with a solution verifier. Automatic tests are then executed to check the validity of the task's implementation.

## 2.2 Task Implementation

Each task in the benchmark comprises of a text file with the task description and an example input/output pair. In addition, each task includes a Python class which implements the following three methods:

- `generate_problem(n, random_seed)`: The **input generator** produces a problem instance parameterized by the problem size $n$ and a random seed. Generators are expected to produce problem instances that take longer to solve as $n$ increases (as in Ziogas et al. (2021)). For example, in QR decomposition the matrix produced by `generate_problem` is of size $n \times n$, and in the graph coloring task, $5n$ is the number of edges in the graph that is produced.

- `solve(problem)`: The **reference solver** takes a generated problem instance and computes its solution. We use this as the performance reference against which speedups are calculated. Whenever possible, we prefer solvers that are direct wrappers of functions from popular open-source Python libraries, such as NumPy, SciPy, CVXPY, sk-learn and NetworkX.

- `is_solution(problem, proposed_solution)`: The **solution verifier** validates an LM-proposed solution for the given problem instance, returning a boolean value.

We automatically validate each contributed AlgoTune task in two ways. First, we confirm that as the input size $n$ grows the solver's runtime increases. Second, we check that for every task, when picking a few input samples from the development set, running the solver with different random seeds still produces outputs accepted by the verifier. We observed this test catching subtle issues in verifiers. For example, we found that in problems that have multiple valid solutions this test can catch verifiers that only accept one possible solution.

Table 1 shows an overview of the categories of tasks in AlgoTune and the packages the reference solvers use. AlgoTune contains a diverse set of tasks covering mathematics, computer science and physics. For illustrative purposes, we list four example tasks here:

- `gzip_compression`: Compress data, such that when decompressed it matches the input data exactly and the compressed data's size is less than or equal to the size of input data when compressed with Python's built-in gzip function.

- `chacha_encryption`: Encrypt a given plaintext using ChaCha20-Poly1305 with a provided key, nonce, and optional associated data (AAD).

- `graph_isomorphism`: Given two isomorphic undirected graphs $G_1$ and $G_2$, find a mapping $f$ between their nodes such that adjacency is preserved. Meaning, if nodes $u$ and $v$ are connected in $G_1$, then nodes $f(u)$ and $f(v)$ must be connected in $G_2$.

- `discrete_log`: Given a prime number $p$, a generator $g$, and a value $h$, compute the discrete logarithm $x$ such that: $g^x \equiv h \pmod{p}$.

**The Importance of the Problem Input Distribution.** The hardness of optimizing each task is dependent not only on the task itself but also on the input data distribution. Therefore, when building the benchmark, we had to focus not only on finding a wide variety of challenging tasks, but also on finding an input problem distribution for each task that is representative of real-world, non-trivial inputs for the solvers. This helps ensure that the LM-synthesized solver is not overly specialized to a narrow set of problems. For example, in the Lasso task (Tibshirani, 1996), a narrowly-defined

Table 1: AlgoTune consists of 154 tasks from 13 categories. This table shows which packages are used in the reference solvers in each category. Note that a reference solver can use multiple packages.

| Category | Task Count | Top 3 Packages Used in Reference Solvers | Example Task |
|---|---|---|---|
| Matrix Operations | 29 | numpy (29), scipy (13), ast (1) | cholesky_factorization |
| Convex Optimization | 28 | numpy (28), cvxpy (23), scipy (2) | aircraft_wing_design |
| Discrete Optimization | 19 | ortools (13), pysat (4), numpy (4) | btsp |
| Graphs | 16 | numpy (14), networkx (9), scipy (5) | articulation_points |
| Signal Processing | 13 | scipy (13), numpy (13) | affine_transform_2d |
| Differential Equation | 12 | scipy (12), numpy (12) | ode_brusselator |
| Statistics | 9 | numpy (9), scipy (6), sklearn (4) | correlate2d_full_fill |
| Nonconvex Optimization | 6 | numpy (6), sklearn (3), hdbscan (1) | clustering_outliers |
| Numerical Methods | 6 | numpy (6), scipy (4) | cumulative_simpson_1d |
| Cryptography | 5 | cryptography (3), hmac (3), sympy (2) | aes_gcm_encryption |
| Computational Geometry | 4 | numpy (4), scipy (3), faiss (1) | convex_hull |
| Control | 4 | numpy (4), cvxpy (2), scipy (2) | feedback_controller_design |
| Misc. | 3 | numpy (3), hmac (1), mpmath (1) | base64_encoding |

data distribution could lead to the optimal regressor to always being the same (for example, with a slope of 1), and the LM-synthesized solver could exploit this by hard-coding the solution to that value rather than solving the problem from scratch.

**The Importance of Writing Complete Tests.**    During development of the AlgoTune infrastructure, we found that sometimes an LM would find what appeared to be substantially more efficient code for a certain task, but upon manual inspection it would turn out to be a solution that passed our tests by finding a loophole and without actually solving the given task. This is analogous to the reward hacking phenomenon observed in reinforcement learning (Amodei et al., 2016). For example, in an initial version of our vector quantization solution verifier, we only tested whether the quantization was valid, but did not check whether the quantization error was optimal. This led to an LM generating a fast, trivial, and suboptimal quantization. We fixed this by adding an optimality test in the solution verifier.

**Do the AlgoTune reference functions have to have "optimal" runtime?**    Benchmarks play a few roles in the current research landscape, including 1) allowing for comparison between different LMs and 2) tracking the progress of the field over time. Both of these are possible with AlgoTune, even if the reference solvers for each task are not "optimal". Since new algorithmic and coding innovations are published constantly, it would be unmanageable to try and maintain a library of 154 solvers that are all state-of-the-art in their domain, both in terms of using the best-known algorithm and in terms of implementing that algorithm in the lowest-level and most efficient way (using Numba or C bindings for Python with Cython). Therefore, we prioritize correctness in our reference implementations, which leads us to source the reference solvers from widely-used Python libraries.

## 2.3  Evaluation Protocol

Agents are allowed to browse online resources while developing code for the tasks in AlgoTune. Agents are also allowed to execute code, and iterate freely on the provided development set of inputs for each task (which, of course, is different from the test inputs that we use to determine the final speedup score on each task). We also allow agents to write code that requires compilation, and we exclude the compilation time from runtime timing measurements (we allow up to 2 minutes of compilation time per task).

**Evaluation.**    For each task, we pick a problem size $n$ such that the task takes our reference solver 100ms to solve on one CPU core, similar to the setup in Ziogas et al. (2021). As in Ziogas et al., due to memory constraints, for a few tasks we use a lower time target (Appendix H reports timing information for each task). To compute a model's score on a given task, we first run its generated code on the test input instances. If the task verifier deems all outputs valid, we assign the model a speedup score: the ratio between reference code runtime and LM code runtime over the test inputs. Solutions that yield invalid outputs or that have a speedup of under $1\times$ are assigned a speedup of $1\times$. The overall AlgoTune score for a specific model is the harmonic mean of its scores on all tasks. We

use the harmonic mean since it is appropriate for averaging speed-up ratios (Smith, 1988; Eeckhout, 2024).

To reliably measure runtime, we do the following in each candidate solver's evaluation: for each problem instance, we first run an untimed warmup run, as is common practice in code benchmarking (Georges et al., 2007; Blackburn et al., 2008). This is then followed by one timed measurement. This is repeated 10 times, of which only the *minimum time* is kept (Arnold et al., 2000). Timing for each run is captured using `time.perf_counter_ns`. We run all measurements on an AMD EPYC 9454 CPU with 14GB of memory.

We note that the choice of problem size $n$ can result in different optimal algorithms, as is the case in matrix multiplication (Smirnov, 2013). Due to budget constraints, we pick a specific problem size for each task, but we emphasize that AlgoTune can operate on arbitrary problem sizes.

## 3 The AlgoTuner Agent

In order to evaluate the frontier LMs on our benchmark, we adopt a SWE-Agent like setup (Yang et al., 2024) wherein the model interacts with a computer environment to iteratively edit code and receive feedback while attempting to optimize an AlgoTune solver.

To get feedback on the current solver performance, we allow the agent to run its code on a development set of inputs for the task, after which we send back timing information and statistics regarding how many instances were solved correctly or timed out. We evaluate all LMs with a fixed budget of $1 for each task. The agent continuously queries the LM to improve its solution until the budget runs out, at which point we submit its best code (as judged by its runtime on the development examples). The score we assign the agent is its speedup over the reference implementation on a held-out test set of inputs for the given task. If this computation results in a score that is less than 1, we assign the agent a mercy score of 1.

In addition to the packages used in the reference solvers, AlgoTune can also use a variety of Python packages, including: Cython (Behnel et al., 2011), Numba (Lam et al., 2015), and Dask (Rocklin, 2015). For a complete list of packages available to the agent, see Appendix D.

We also implement commands for the agent that are specific to code optimization: `profile` allows the agent to use a profiler to see which parts of the code take the most time, and `reference` runs the reference solver on a given input and reports back the output and runtime. See Appendix B for further details of the agent setup, prompt, and commands.

## 4 Results

We run AlgoTuner using four frontier LMs through the LiteLLM library (BerriAI, 2025): `o4-mini-high` (OpenAI, 2025), `Claude Opus 4 20250514` (Anthropic, 2025b), `Gemini 2.5 Pro` (Google DeepMind, 2025), and `DeepSeek R1 0528` (Guo et al., 2025). In all cases where applicable, the highest thinking setting was used. We present our results in Table 2, showing that models are able to optimize our reference solvers somewhat, with o4-mini achieving an AlgoTune score of $1.72\times$, which is the harmonic mean of its speedup ratios across all tasks, with tasks for which the model cannot find a quicker solver given a speedup of $1\times$. Our analysis shows that these speedups are mostly surface level, simple optimizations, and do not represent any algorithmic innovations.

Table 2: AlgoTuner scores for each LM, with a budget of $1 for each task. Speedup is calculated as the harmonic mean of the speedups across tasks.

|  | o4-mini | R1 | Gemini 2.5 Pro | Claude Opus 4 |
|---|---|---|---|---|
| AlgoTune Score | $1.72\times$ | $1.70\times$ | $1.51\times$ | $1.33\times$ |

### 4.1 Quantitative Analysis

Figure 3 shows how agent-generated code scores on AlgoTune (on the development set of input problems) at intermediate budget checkpoints. Both o4-mini-high and R1 achieve better scores after

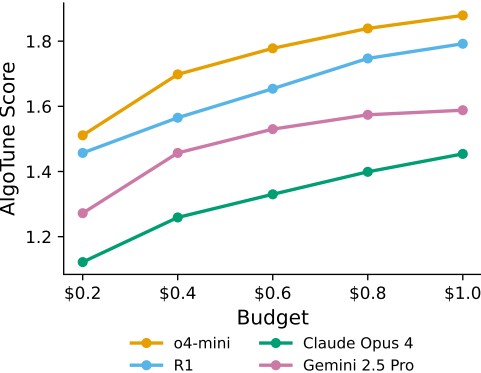

Figure 3: AlgoTune scores (on the development set of input problems) across all tasks, during the running of AlgoTuner, for intermediate budget splits, up to the total budget of $1.

Table 3: The top packages added or removed by o4-mini's optimized solvers (compared to those used by the reference solvers), across all 92 tasks it sped up, ranked by absolute change.

| Package | Reference | LM Generated | Δ |
|---|---|---|---|
| numba | 1 | 25 | +24 |
| scipy | 61 | 67 | +6 |
| ecos | 0 | 2 | +2 |
| faiss | 2 | 4 | +2 |
| pysat | 4 | 1 | -3 |
| hmac | 4 | 0 | -4 |
| sklearn | 9 | 5 | -4 |
| networkx | 12 | 2 | -10 |
| numpy | 132 | 118 | -14 |
| cvxpy | 27 | 10 | -17 |

spending $0.1, than Claude Opus 4 and Gemini 2.5 Pro achieve after spending the full budget. In Table 3, we show packages used by o4-mini-high on the 59.7% of tasks where it managed to get a speedup of at least $1.1\times$, and how those packages differ from the packages used by the reference solvers. We can see that it frequently used Numba to write efficient solvers, and that it frequently rewrote solvers that used NetworkX, CVXPY and OR-Tools to remove those dependencies. For additional quantitative analysis, see §C.

## 4.2 Qualitative Analysis

Manually reviewing AlgoTuner's synthesized solvers shows that its optimizations are mostly surface-level. We go over a few of the different optimization patterns used by AlgoTuner below:

**Using a Better Implementation.** In several tasks, AlgoTuner replaces the reference implementation with a call to a specialized, more efficient function. For example, in `feedback_controller_design`, instead of using CVXPY, a call to `scipy.linalg.solve_discrete_are` is made (see Fig. 4), and in the `lyapunov_stability` task, instead of CVXPY the optimized code calls `scipy.linalg.solve_discrete_lyapunov`.

**Better Library Usage.** In several tasks, the default usage of a library (e.g., operations or parameters of a function call) are swapped for more optimized ones in the optimized code. This is done without significantly changing the algorithmic structure of the solution. In `psd_cone_projection`, the optimized code uses the same level of abstraction as the reference, but with more efficient usage of NumPy (see Fig. 5).

**Rewriting Using Low-Level Operations.** In `graph_isomorphism`, instead of using NetworkX objects, AlgoTuner's code works on adjacency lists and runs a single Weisfeiler–Lehman (Shervashidze et al., 2011) pass, resulting in far fewer recursive calls than the NetworkX implementation. This results in a 52x speedup. In `communicability`, AlgoTuner's solver uses BLAS operations instead of pure Python, which leads to a speedup of more than 142x. In `ode_hodgkinhuxley`, AlgoTuner's code uses a Numba kernel instead of the reference's SciPy call, achieving a 112x speedup. In another case, the reference solution uses SciPy's `stats.wasserstein_distance`, which incurs Python overhead and extra work each call; AlgoTuner's code compiles a Numba kernel that streams once over the data, so after a one-time JIT compilation it runs at near-C speed, leading to a speedup of more than 4x (see Fig. 6).

**Failure Analysis.** We next present a few examples of AlgoTuner failing to optimize code.

In the `svm` task, our reference solver formulates the SVM as a convex program and then solves it using CVXPY. AlgoTuner with o4-mini, Claude Opus 4, and R1 reach the same implementation,

```
import cvxpy as cp                          from scipy.linalg import
                                               solve_discrete_are
def solve(A, B):
  n, m = A.shape[0], B.shape[1]            def solve(A, B):
  Q = cp.Variable((n, n), symmetric=         n, m = A.shape[0], B.shape[1]
    True)                                     Q = np.eye(n)
  L = cp.Variable((m, n))                     R = np.eye(m)
  cons = [                                    P = solve_discrete_are(A, B, Q, R)
    cp.bmat([                                 PB = P.dot(B)
      [Q, Q @ A.T + L.T @ B.T],               S = R + PB.T.dot(B)
      [A @ Q + B @ L, Q]                      N = PB.T.dot(A)
    ]) >> np.eye(2 * n),                      K = -np.linalg.solve(S, N)
    Q >> np.eye(n),                           return P, K
  ]
  obj = cp.Minimize(0)
  prob = cp.Problem(obj, cons)
  prob.solve()
  K = L.value @ np.linalg.inv(Q.value)
  P = np.linalg.inv(Q.value)
  return P, K
```

Figure 4: **Left:** Our feedback controller task starts with a reference CVXPY implementation solving an SDP formulation. **Right:** AlgoTuner with o4-mini improves upon the runtime by a factor of 81 by rewriting it to use SciPy's discrete algebraic Ricatti equation (DARE) solver.

```
def solve(A):                              def solve(A):
  eigvals, eigvecs = np.linalg.eig(A)        eigvals, eigvecs = np.linalg.eigh(A)
  eigvals = np.maximum(eigvals, 0)           eigvals[eigvals < 0] = 0
  E = np.diag(eigvals)                       X = (eigvecs * eigvals) @ eigvecs.T
  X = eigvecs @ E @ eigvecs.T                return X
  return X
```

Figure 5: **Left:** Our original code for a PSD cone projection of a symmetric matrix projects the eigenvalues to be non-negative. **Right:** AlgoTuner with Claude Opus 4 improves the code by a factor of 8 by 1) using a symmetric eigendecomposition, and 2) not forming the eigenvalue matrix and instead applying them directly to the eigenvectors.

which results in no speedup. Gemini 2.5 Pro is not able to come up with a solver that produces valid results for every test instance.

In the `lasso` task, our reference solver uses scikit-learn's optimized `linear_model.Lasso`, while AlgoTuner with Claude Opus 4 wrote a Lasso regressor using pure Python and Numba. This resulted in code that ran at 0.33x the time of the reference solver, and the agent was unable to improve on this due to reaching the budget limit.

## 5   Related Work

Program synthesis, the automatic generation of programs subject to input constraints, is a long-standing problem in computer science and has been previously referred to as a "holy grail" of the field (Gulwani et al., 2017). Predating modern language models, a variety of approaches have been applied to the problem including constraint satisfaction (Torlak and Bodík, 2013; Solar-Lezama, 2008), statistical methods (Raychev et al., 2014), and enumerative search (Alur et al., 2015). We especially note that Massalin (1987) introduced the concept of "superoptimization," the problem of finding the fastest possible compilation of source program to a target language. We direct readers to Gulwani et al. (2017); David and Kroening (2017) for a general survey and Allamanis et al. (2018) for machine learning methods specifically.

Recent benchmarks challenge LMs in real-world problem solving, from fixing software bugs to answering medical questions (Jimenez et al., 2024; Arora et al., 2025). Prior work using LMs for code generation has focused on challenging the LMs to program specific functions, measuring only correctness but not speed (Chen et al., 2021; Nijkamp et al., 2022; Li et al., 2022; Fried et al., 2022;

```python
from scipy.stats import
    wasserstein_distance

def solve(u, v):
  domain = list(range(1, u.shape[0]+1))
  return wasserstein_distance(
    domain, domain, u, v)
```

```python
@numba.njit(cache=True, fastmath=True)
def wass(u,v):
    cumulative_diff, total_distance =
    0.0, 0.0
    for i in range(n - 1):
        cumulative_diff += u[i] - v[i]
        total_distance += abs(
    cumulative_diff)
    return total_distance

def solve(u, v):
  return wass(u, v)
```

Figure 6: **Left:** Our reference implementation for the 1D Wasserstein task calls into SciPy's function. **Right:** AlgoTuner with Gemini 2.5 Pro improves the performance by a factor of 4 by writing Numba-jitted code for the difference between the CDFs of the distributions.

Cassano et al., 2023; Li et al., 2023; Liu et al., 2023; Tian et al., 2024; Jiang et al., 2024). In AlgoTune we check for correctness but score models based on the speed of their generated code.

Recent works (Shypula et al., 2023; Qiu et al., 2024; Huang et al., 2024; Coignion et al., 2024; Waghjale et al., 2024; Du et al., 2024) have proposed benchmarks that challenge LMs to make LeetCode and coding competition-style code more efficient. However, the tasks in AlgoTune provide a wider and more realistic range of optimization challenges. In the future, optimizations found on AlgoTune could directly lead to functions in popular open source projects such as Numpy, SciPy and NetworkX becoming faster.

Misra (2024) showed that current state-of-the-art LMs struggle to optimize code, echoing findings from (Qiu et al., 2024; Huang et al., 2024). Concurrently with our work, Ouyang et al. (2025) task LMs with optimizing GPU kernel code for neural networks. AlgoTune contains tasks in many varied domains and does not just focus on deep learning functions. Other concurrent works, including Fan et al. (2023); Romera-Paredes et al. (2024); Imajuku et al. (2025); Chen et al. (2025); Sun et al. (2025); Novikov et al. (2025); Tang et al. (2025) present benchmarks and AI systems that optimize the solution quality of problems in computer science and math (for example, finding the shortest traveling salesman path in a graph). In AlgoTune, we instead focus on making functions run *faster* while meeting a solution quality threshold. AlgoTune also consists of a wider variety of problems, including popular algorithms that have not been previously used in benchmarks, such as those for encryption (AES GSM, ChaCha), compression (gzip) and hashing (SHA-256). A final concurrent work, Shetty et al. (2025), challenges LMs to optimize the runtime of functions in popular Python repositories including NumPy, pandas and Pydantic. Our approach complements their work since we focus on implementing end-to-end algorithms from a wide variety of domains, whereas their benchmark focuses on isolated functions from machine learning, data science and image processing repositories.

In addition to being used to directly write efficient code, LMs are also being used to reduce compiled code size (Italiano and Cummins, 2024), and in general to make compilers more efficient (Cummins et al., 2024a;b). This provides an orthogonal approach, which in the future could be combined with AlgoTuner-style optimizations to lead to further speedups.

Agents also struggle in tasks outside the code debugging and efficiency domains. ScienceAgent-Bench (Chen et al., 2024) showed that agents struggle with data-analysis and simulation challenges. ML engineering benchmarks like ML-Dev-Bench (Padigela et al., 2025) and MLE-bench (Chan et al., 2024) demonstrate that agents struggle with complex debugging and creative model improvements, rarely matching expert performance.

# 6 Limitations and Future Work

Our benchmark consists of algorithmic tasks that have well-defined solvers, input data generators, and solution verifiers, which must be written and checked manually. Future work could enable optimizing systems for which data generators and verifiers are hard or impossible to write, such as a web server or an operating system.

As mentioned in §2, we verify solutions on a fixed set of test inputs. Shortcomings in the test input distribution or in verifier completeness leave open the possibility for incorrect LM-synthesized code that passes tests. Future work could use formal verification to better verify LM-synthesized code, but this introduces constraints on programming language applicability and in some cases may require proof construction. In our experience, manual inspection of LM-generated code was sufficient to identify such potential issues.

## 7 Conclusion

AlgoTune introduces a benchmark where LMs are tasked with optimizing code in a wide range of domains. Along with a comprehensive benchmark suite consisting of 154 tasks, AlgoTune provides infrastructure for verifying and timing LM-generated candidate solutions, which makes arbitrary numerical function optimization streamlined and easy to deploy.

We showed that our AlgoTuner agent can frequently optimize the functions in AlgoTune. Paired with four frontier LMs, AlgoTuner is able to provide surface level optimizations but is not able to come up with novel algorithms. Future work could deploy AlgoTune on other platforms, including GPUs/TPUs and distributed systems, with only minimal code changes. We hope that our work leads to LM systems that enable development of highly efficient code in a wide variety of domains, leading to scientific progress.

## Acknowledgments

We thank David Alvarez-Melis and Matthias Kümmerer for their valuable feedback and fruitful discussions. Meta was involved only in an advisory role. All experimentation and data processing, including the use of open source models and datasets, was conducted at the University of Tübingen and Princeton University. The authors acknowledge support by the state of Baden-Württemberg through bwHPC. The authors thank the International Max Planck Research School for Intelligent Systems (IMPRS-IS) for supporting Ori Press and Nathanael Bosch. MB is a member of the Machine Learning Cluster of Excellence, funded by the Deutsche Forschungsgemeinschaft (DFG, German Research Foundation) under Germany's Excellence Strategy – EXC number 2064/1 – Project number 390727645 and acknowledges support by the German Research Foundation (DFG): SFB 1233, Robust Vision: Inference Principles and Neural Mechanisms, TP 4, Project No: 276693517. He further acknowledges financial support from Open Philanthropy Foundation funded by the Good Ventures Foundation. This work was supported by the Tübingen AI Center.

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

# A Results

In Table 4, we show a summary of the results of AlgoTuner for each of the four frontier models tests. In Table 5, we detail the per-task timings for every model and task.

Table 4: AlgoTuner speedup when using each LM, with a budget of $1 for each task. Speedup percentage is calculated as the percentage of tasks for which AlgoTuner gets at least a $1.1\times$ speedup.

|  | R1 | o4-mini | Gemini 2.5 Pro | Claude Opus 4 |
|---|---|---|---|---|
| Pct. of Tasks Sped Up | 61.0% | 59.7% | 49.4% | 40.3% |

Table 5: Per task speedup for AlgoTuner, using four frontier LMs. Speedup is calculated as the ratio between the reference solve function's time and the LM-generated solve function's time.

| Task | o4-mini | R1 | Gemini 2.5 Pro | Claude Opus 4 | Claude Opus 4 |
|---|---|---|---|---|---|
| aes_gcm_encryption | 1.05 | 1.03 | 1.00 | 1.54 | 1.00 |
| affine_transform_2d | 1.00 | 1.00 | 1.00 | 1.00 | 1.00 |
| aircraft_wing_design | 1.00 | 1.00 | 1.70 | 1.02 | 1.03 |
| articulation_points | 4.91 | 5.93 | 1.00 | 3.22 | 3.13 |
| base64_encoding | 1.00 | 1.00 | 1.00 | 1.30 | 1.00 |
| battery_scheduling | 27.48 | 13.18 | 26.28 | 12.84 | 20.01 |
| btsp | 1.00 | 2.76 | 1.62 | 1.00 | 1.00 |
| capacitated_facility_location | 8.23 | 16.99 | 8.53 | 7.47 | 1.00 |
| chacha_encryption | 1.53 | 1.04 | 1.00 | 1.29 | 1.00 |
| channel_capacity | 1.00 | 1.07 | 1.19 | 1.15 | 1.05 |
| chebyshev_center | 5.65 | 3.69 | 4.91 | 4.87 | 4.70 |
| cholesky_factorization | 1.12 | 1.00 | 1.00 | 1.10 | 1.00 |
| clustering_outliers | 1.32 | 1.16 | 1.00 | 1.16 | 1.13 |
| communicability | 59.76 | 66.39 | 197.67 | 53.17 | 106.19 |
| convex_hull | 1.00 | 5.09 | 4.95 | 1.00 | 1.00 |
| convolve2d_full_fill | 161.95 | 155.32 | 175.96 | 145.48 | 140.17 |
| convolve_1d | 1.05 | 1.06 | 1.03 | 1.02 | 1.00 |
| correlate2d_full_fill | 123.88 | 177.11 | 129.27 | 64.59 | 128.44 |
| correlate_1d | 1.04 | 1.03 | 1.09 | 1.06 | 1.00 |
| count_connected_components | 4.21 | 6.04 | 2.61 | 3.15 | 4.01 |
| count_riemann_zeta_zeros | 1.00 | 1.00 | 1.00 | 1.00 | 1.00 |
| cumulative_simpson_1d | 14.64 | 12.82 | 6.99 | 9.26 | 1.00 |
| cumulative_simpson_multid | 1.00 | 1.00 | 1.00 | 1.13 | 1.00 |
| cvar_projection | 1.90 | 2.79 | 1.00 | 1.00 | 1.72 |
| cyclic_independent_set | 1.00 | 39.92 | 1.00 | 1.00 | 1.01 |
| dct_type_I_scipy_fftpack | 1.07 | 1.01 | 1.21 | 1.00 | 1.00 |
| delaunay | 3.55 | 3.75 | 1.00 | 1.73 | 1.00 |
| dijkstra_from_indices | 1.00 | 1.00 | 1.00 | 1.00 | 1.00 |
| discrete_log | 1.00 | 1.33 | 1.00 | 1.00 | 1.00 |
| dst_type_II_scipy_fftpack | 1.85 | 1.47 | 1.00 | 1.00 | 1.00 |
| dynamic_assortment_planning | 218.65 | 48.51 | 33.70 | 7.73 | 1.51 |
| earth_movers_distance | 1.02 | 1.00 | 1.00 | 1.00 | 1.00 |
| edge_expansion | 26.62 | 28.80 | 1.00 | 1.00 | 1.06 |
| eigenvalues_complex | 1.48 | 1.46 | 1.45 | 1.49 | 1.44 |
| eigenvalues_real | 2.47 | 2.52 | 2.42 | 2.51 | 2.42 |
| eigenvectors_complex | 1.04 | 1.02 | 1.01 | 1.01 | 1.00 |
| eigenvectors_real | 1.02 | 1.01 | 1.04 | 1.01 | 1.01 |
| elementwise_integration | 1.00 | 1.00 | 1.00 | 1.00 | 1.00 |
| feedback_controller_design | 343.02 | 334.31 | 77.08 | 1.00 | 1.04 |
| fft_cmplx_scipy_fftpack | 2.38 | 2.36 | 1.43 | 2.35 | 2.58 |
| fft_convolution | 1.00 | 1.00 | 1.00 | 1.00 | 1.00 |
| fft_real_scipy_fftpack | 1.00 | 1.40 | 1.06 | 1.14 | 4.10 |

| Task | o4-mini | R1 | Gemini 2.5 Pro | Claude Opus 4 | Claude Opus 4 |
|---|---|---|---|---|---|
| firls | 1.00 | 1.00 | 1.00 | 1.00 | 1.01 |
| generalized_eigenvalues_complex | 3.65 | 5.26 | 1.96 | 5.39 | 3.49 |
| generalized_eigenvalues_real | 2.42 | 3.13 | 2.27 | 2.46 | 2.39 |
| generalized_eigenvectors_complex | 2.45 | 3.36 | 2.70 | 1.11 | 1.02 |
| generalized_eigenvectors_real | 1.43 | 1.68 | 3.19 | 1.36 | 1.92 |
| graph_coloring_assign | 42.88 | 1.48 | 1.10 | 1.19 | 1.00 |
| graph_global_efficiency | 1.07 | 15.65 | 16.61 | 15.85 | 14.19 |
| graph_isomorphism | 40.51 | 75.81 | 50.35 | 80.10 | 27.41 |
| graph_laplacian | 1.00 | 1.00 | 1.00 | 1.00 | 1.00 |
| group_lasso | 1.00 | 1.00 | 1.00 | 1.00 | 1.00 |
| gzip_compression | 1.34 | 1.00 | 1.00 | 1.00 | 1.00 |
| integer_factorization | 1.00 | 1.00 | 1.00 | 1.00 | 1.00 |
| job_shop_scheduling | 1.77 | 1.81 | 1.37 | 1.61 | 1.05 |
| kalman_filter | 46.98 | 15.76 | 9.93 | 1.00 | 1.00 |
| kcenters | 2.57 | 1.86 | 1.00 | 1.00 | 1.00 |
| kd_tree | 1.13 | 1.02 | 1.00 | 1.01 | 1.00 |
| kernel_density_estimation | 1.00 | 1.00 | 1.00 | 1.00 | 1.00 |
| kmeans | 16.87 | 12.53 | 15.25 | 9.29 | 15.49 |
| ks_test_2samp | 1.00 | 1.00 | 1.00 | 1.11 | 1.00 |
| l0_pruning | 1.00 | 1.38 | 2.48 | 1.42 | 2.71 |
| l1_pruning | 17.69 | 1.85 | 1.79 | 1.29 | 1.39 |
| lasso | 1.18 | 1.57 | 1.00 | 1.00 | 1.00 |
| least_squares | 1.00 | 2.32 | 1.33 | 1.09 | 2.02 |
| linear_system_solver | 1.12 | 1.07 | 1.09 | 1.11 | 1.04 |
| lp_box | 13.32 | 17.01 | 15.26 | 14.36 | 13.33 |
| lp_centering | 1.01 | 1.01 | 1.00 | 1.00 | 1.00 |
| lp_mdp | 865.71 | 369.78 | 327.67 | 61.69 | 7.22 |
| lqr | 1.00 | 1.25 | 1.25 | 1.09 | 1.00 |
| lti_simulation | 1.15 | 16.39 | 2.05 | 1.00 | 1.00 |
| lu_factorization | 1.00 | 1.00 | 1.00 | 1.01 | 1.20 |
| lyapunov_stability | 142.10 | 189.60 | 82.78 | 107.65 | 118.83 |
| markowitz | 1.00 | 1.00 | 1.00 | 1.04 | 1.00 |
| matrix_completion | 1.00 | 1.00 | 1.00 | 1.01 | 1.02 |
| matrix_exponential | 1.00 | 1.00 | 1.00 | 1.00 | 1.00 |
| matrix_exponential_sparse | 1.00 | 1.00 | 1.00 | 1.00 | 1.00 |
| matrix_multiplication | 1.06 | 1.00 | 1.00 | 1.00 | 1.00 |
| matrix_sqrt | 1.04 | 1.00 | 1.00 | 1.00 | 1.00 |
| max_clique_cpsat | 28.05 | 13.22 | 9.34 | 1.00 | 5.41 |
| max_common_subgraph | 46.79 | 2.15 | 9.09 | 1.60 | 1.28 |
| max_flow_min_cost | 3.03 | 8.81 | 1.63 | 9.18 | 14.34 |
| max_independent_set_cpsat | 76.14 | 1.68 | 1.30 | 1.61 | 1.00 |
| max_weighted_independent_set | 1.55 | 1.00 | 1.26 | 1.00 | 1.00 |
| min_dominating_set | 1.00 | 1.85 | 1.00 | 1.00 | 1.57 |
| min_weight_assignment | 1.00 | 1.70 | 1.00 | 1.56 | 1.01 |
| minimum_spanning_tree | 9.90 | 9.09 | 1.00 | 9.72 | 1.00 |
| minimum_volume_ellipsoid | 6.65 | 45.38 | 16.00 | 1.01 | 1.00 |
| multi_dim_knapsack | 2.23 | 56.93 | 5.50 | 2.15 | 1.49 |
| nmf | 1.00 | 1.16 | 1.03 | 1.00 | 1.22 |
| ode_brusselator | 301.75 | 1.62 | 1.00 | 1.00 | 1.00 |
| ode_fitzhughnagumo | 1.03 | 1.08 | 1.00 | 1.12 | 1.00 |
| ode_hires | 8.55 | 29.24 | 25.75 | 8.14 | 3.84 |
| ode_hodgkinhuxley | 165.94 | 52.40 | 112.08 | 5.18 | 5.50 |
| ode_lorenz96_nonchaotic | 1.67 | 2.86 | 1.78 | 1.66 | 1.70 |
| ode_lotkavolterra | 814.44 | 5.09 | 53.56 | 1.00 | 2.17 |
| ode_nbodyproblem | 54.21 | 50.07 | 17.31 | 50.61 | 17.02 |
| ode_seirs | 3084.39 | 1.00 | 43.75 | 1.64 | 13.04 |
| ode_stiff_robertson | 12.01 | 68.88 | 2.25 | 2.73 | 2.23 |
| ode_stiff_vanderpol | 2062.53 | 90.93 | 1.00 | 2.32 | 1.86 |
| odr | 1.01 | 1.01 | 1.00 | 1.00 | 1.00 |
| optimal_advertising | 1.00 | 1.29 | 1.00 | 1.00 | 1.00 |
| outer_product | 1.00 | 1.02 | 1.00 | 1.78 | 1.00 |

| Task | o4-mini | R1 | Gemini 2.5 Pro | Claude Opus 4 | Claude Opus 4 |
|---|---|---|---|---|---|
| pagerank | 1.01 | 1.04 | 30.97 | 1.00 | 4.22 |
| pca | 3.62 | 4.15 | 2.16 | 2.41 | 1.00 |
| pde_burgers1d | 1.02 | 4.17 | 3.43 | 1.00 | 4.03 |
| pde_heat1d | 1.80 | 1.92 | 1.00 | 1.00 | 1.00 |
| polynomial_mixed | 99.78 | 4.32 | 1.00 | 1.05 | 1.01 |
| polynomial_real | 73.71 | 134.71 | 1.00 | 1.00 | 1.00 |
| power_control | 304.84 | 346.26 | 160.39 | 1.00 | 17.67 |
| procrustes | 2.32 | 1.03 | 1.00 | 1.01 | 1.00 |
| psd_cone_projection | 8.96 | 2.88 | 8.11 | 2.46 | 8.46 |
| qp | 1.44 | 1.64 | 1.68 | 1.74 | 1.70 |
| qr_factorization | 7.95 | 1.00 | 1.08 | 1.01 | 1.17 |
| quantile_regression | 1.17 | 1.18 | 1.41 | 1.17 | 1.15 |
| queens_with_obstacles | 2.50 | 3.00 | 2.87 | 1.04 | 1.73 |
| queuing | 1.10 | 1.00 | 1.00 | 1.09 | 1.00 |
| qz_factorization | 1.73 | 1.00 | 1.00 | 1.00 | 1.00 |
| randomized_svd | 3.79 | 4.51 | 1.00 | 2.49 | 1.00 |
| rbf_interpolation | 1.02 | 1.00 | 1.00 | 1.00 | 1.00 |
| rectanglepacking | 1.84 | 1.00 | 2.29 | 1.00 | 1.00 |
| robust_kalman_filter | 7.05 | 2.06 | 8.63 | 1.01 | 3.47 |
| robust_linear_program | 6.49 | 1.00 | 6.51 | 1.00 | 1.06 |
| rocket_landing_optimization | 1.00 | 1.63 | 1.03 | 1.00 | 1.00 |
| rotate_2d | 1.00 | 1.00 | 1.00 | 1.00 | 1.00 |
| set_cover | 29.74 | 6.70 | 1.79 | 1.71 | 1.00 |
| set_cover_conflicts | 5.59 | 1.96 | 2.07 | 1.00 | 2.18 |
| sha256_hashing | 1.00 | 1.00 | 1.00 | 1.00 | 1.00 |
| shift_2d | 1.00 | 1.00 | 1.00 | 1.00 | 1.00 |
| shortest_path_dijkstra | 2.18 | 2.33 | 2.44 | 1.97 | 1.00 |
| sinkhorn | 1.62 | 1.86 | 2.23 | 1.00 | 1.00 |
| sparse_eigenvectors_complex | 1.00 | 1.00 | 1.00 | 1.00 | 1.00 |
| sparse_lowest_eigenvalues_posdef | 1.89 | 1.83 | 1.63 | 1.50 | 1.74 |
| sparse_lowest_eigenvectors_posdef | 1.88 | 2.47 | 1.74 | 1.49 | 1.50 |
| sparse_pca | 4.91 | 9.08 | 5.44 | 1.75 | 1.61 |
| spectral_clustering | 4.61 | 13.51 | 9.88 | 10.53 | 1.00 |
| stable_matching | 1.73 | 1.54 | 1.58 | 1.30 | 1.49 |
| svd | 1.62 | 1.02 | 1.00 | 1.00 | 1.00 |
| svm | 1.00 | 1.00 | 1.00 | 1.00 | 1.00 |
| sylvester_solver | 1.03 | 1.00 | 1.00 | 1.00 | 1.00 |
| tensor_completion_3d | 203.38 | 24.61 | 33.87 | 2.52 | 2.49 |
| toeplitz_solver | 1.00 | 1.00 | 1.00 | 1.00 | 1.00 |
| tsp | 1.00 | 1.32 | 1.00 | 1.81 | 1.17 |
| two_eigenvalues_around_0 | 1.92 | 1.70 | 1.75 | 1.80 | 1.67 |
| unit_simplex_projection | 3.61 | 3.53 | 1.00 | 1.09 | 1.00 |
| upfirdn1d | 1.00 | 1.13 | 1.00 | 1.00 | 1.00 |
| vector_quantization | 1.00 | 1.00 | 1.00 | 1.00 | 1.01 |
| vectorized_newton | 1.00 | 1.00 | 1.00 | 1.00 | 1.00 |
| vehicle_routing | 1.21 | 1.22 | 2.76 | 1.00 | 1.40 |
| vertex_cover | 1.84 | 1.26 | 2.55 | 1.08 | 1.00 |
| voronoi_diagram | 9.28 | 1.09 | 3.35 | 1.00 | 2.27 |
| wasserstein_dist | 9.82 | 9.87 | 4.66 | 9.56 | 4.58 |
| water_filling | 514.52 | 86.16 | 213.25 | 84.57 | 183.87 |
| zoom_2d | 1.00 | 1.00 | 1.00 | 1.00 | 1.00 |

# B  AlgoTuner Agent Setup

**Initial Prompt.**    The LM receives an initial message, consisting of general instructions on how to use the system (see §B.1), Numba (Lam et al., 2015), Dask (Rocklin, 2015), and Cython (Behnel et al., 2011) (for a full list see Appendix D). Additionally, the LM is given the task's description, which includes input and output descriptions and examples, as well as the task's `solve` and `is_solution` functions. In essence, everything apart from the problem generating function is shown to the LM.

**Using the Interface.**    The goal of the LM is to write a `Solver` class with a `solve()` function that takes problem instances and produces a correct output. To do this, the LM sends messages that must consist of exactly one thought and one command (Yao et al., 2023). Responses given by the system always start with a budget status, for example: `You have so far sent 3 messages, and used up $0.08.  You have $0.92 remaining`. We use the LiteLLM (BerriAI, 2025) API to access all models used. Each model is limited to a budget of $1 per task, and is continuously prompted until its budget runs out. The budget includes both input and output tokens. Where applicable, we set the temperature to $0$ and the `top_p` parameter to $0.95$.

**Commands.**    We detail the available system commands in Table 6. Following Yang et al. (2024) after an edit command is used the modified code is ran through a linter. If the linter raises errors, the code is reverted, and the linter errors are sent back to the LM. When there are no linter errors, the code is evaluated on 100 training samples, with results sent back to the LM. When there are runtime errors, those are also sent back to the LM. When there are no runtime errors, the performance score, along with average evaluation time is reported back to the LM. If the performance score reached is better than any score previously reached, the code state is saved.

Table 6: Available interface commands.

| Command | Description |
| --- | --- |
| edit | Replace a range of lines in a file with new content. Can create new files. |
| delete | Remove a range of lines from a file. |
| ls | List all files in the current working directory. |
| view_file | Display 100 lines of a file from a specified start line. |
| revert | Revert all files to the best-performing version of the code. |
| reference | Get the reference solve's solution for a given input |
| eval | Evaluate the current solve function on the 100 training instances and report results. |
| eval_input | Run the solver on a given input and compare with the oracle. |
| profile | Profile the performance of the solve method on a given input. |
| profile_lines | Profile specified lines on a given input. |

**Message History.**    To manage conversational context within token limits, we truncate the messages send to the LM in the following manner: The initial system prompt and the full content of the most recent five user and five assistant turns are always sent, following (Yang et al., 2024). Messages older than these are truncated to the first 100 characters. If the total token count still exceeds the model's limit, these older, content-truncated messages are progressively dropped starting from the oldest and are replaced by a single placeholder message indicating the truncation is inserted after the system prompt to signal the discontinuity.

## B.1 Initial Prompt

We show the initial prompt given to the language model. The prompt we use is loosely modeled after the one used in SWE-Agent (Yang et al., 2024). For each task, we add a task description and the reference solver implementation (see below).

```
SETTING:
You're an autonomous programmer tasked with solving a specific problem.
    You are to use the commands defined below to accomplish this task.
    Every message you send incurs a cost--you will be informed of your
    usage and remaining budget by the system.
You will be evaluated based on the best-performing piece of code you
    produce, even if the final code doesn't work or compile (as long as
    it worked at some point and achieved a score, you will be eligible).
Apart from the default Python packages, you have access to the following
    additional packages:
 - cryptography
 - cvxpy
 - cython
 - dask
 - diffrax
 - ecos
 - faiss-cpu
 - hdbscan
 - highspy
 - jax
 - networkx
 - numba
 - numpy
 - ortools
 - pandas
 - pot
 - pulp
 - pyomo
 - python-sat
 - scikit-learn
 - scipy
 - sympy
 - torch

YOUR TASK:
Your objective is to define a class named 'Solver' in 'solver.py' with a
    method:
'''
class Solver:
    def solve(self, problem, **kwargs) -> Any:
        """Your implementation goes here."""
        ...
'''

IMPORTANT: Compilation time of your init function will not count towards
    your function's runtime.

This 'solve' function will be the entrypoint called by the evaluation
    harness. Strive to align your class and method implementation as
    closely as possible with the desired performance criteria.
For each instance, your function can run for at most 10x the baseline
    runtime for that instance. Strive to have your implementation run as
    fast as possible, while returning the same output as the baseline
```

function (for the same given input). Be creative and optimize your approach!

Your messages should include a short thought about what you should do, followed by a _SINGLE_ command. The command must be enclosed within ''' and ''', like so:
<Reasoning behind executing the command>
'''
<command>
'''

IMPORTANT: Each set of triple backticks (''') must always be on their own line, without any other words or anything else on that line.

Here are the commands available to you. Ensure you include one and only one of the following commands in each of your responses:
- 'edit': Replace a range of lines with new content in a file. This is how you can create files: if the file does not exist, it will be created. Here is an example:
  '''
  edit
  file: <file_name>
  lines: <start_line>-<end_line>
  ---
  <new_content>
  ---
  '''

  The command will:
  1. Delete the lines from <start_line> to <end_line> (inclusive)
  2. Insert <new_content> starting at <start_line>
  3. If both <start_line> and <end_line> are 0, <new_content> will be prepended to the file

  Example:
  edit
  file: solver.py
  lines: 5-7
  ---
  def improved_function():
      print("Optimized solution")
  ---
- 'ls': List all files in the current working directory.
- 'view_file <file_name> [start_line]': Display 100 lines of '<file_name>' starting from 'start_line' (defaults to line 1).
- 'revert': Revert the code to the best-performing version thus far.
- 'baseline <string>': Query the baseline solver with a problem and receive its solution. If the problem's input is a list, this command would look like:
  '''
  baseline [1,2,3,4]
  '''
- 'eval_input <string>': Run your current solver implementation on the given input. This is the only command that shows stdout from your solver along with both solutions. Example:
  '''
  eval_input [1,2,3,4]
  '''
- 'eval': Run evaluation on the current solution and report the results.

- 'delete': Delete a range of lines from a file using the format:
  ```
  delete
  file: <file_name>
  lines: <start_line>-<end_line>

  The command will delete the lines from <start_line> to <end_line> (
    inclusive)

  Example:
  delete
  file: solver.py
  lines: 5-10
  ```
- 'profile <filename.py> <input>': Profile your currently loaded solve
    method's performance on a given input. Shows the 25 most time-
    consuming lines. Requires specifying a python file (e.g., 'solver.py
    ') for validation, though profiling runs on the current in-memory
    code.
  Example:
  ```
  profile solver.py [1, 2, 3]
  ```

- 'profile_lines <filename.py> <line_number1, line_number2, ...> <input
    >': Profiles the chosen lines of the currently loaded code on the
    given input. Requires specifying a python file for validation.
  Example:
  ```
  profile_lines solver.py 1,2,3 [1, 2, 3]
  ```

**TIPS:**
After each edit, a linter will automatically run to ensure code quality.
    If there are critical linter errors, your changes will not be applied,
     and you will receive the linter's error message. Typically, linter
    errors arise from issues like improper indentation--ensure your edits
     maintain proper code formatting.
**Cython Compilation:** Edits creating or modifying Cython ('.pyx') files
     will automatically trigger a compilation attempt (requires a 'setup.
    py'). You will be notified if compilation succeeds or fails. If it
    fails, the edit to the '.pyx' file will be automatically reverted.
You have 8 CPU cores at your disposal, you can use them for
    parallelization.
If the code runs successfully without errors, the in-memory 'last known
    good code' will be updated to the new version. Following successful
    edits, you will receive a summary of your 'solve' function's
    performance compared to the baseline.
If you get stuck, try reverting your code and restarting your train of
    thought.
Do not put an if __name__ == "__main__": block in your code, as it will
    not be ran (only the solve function will).
Keep trying to better your code until you run out of money. Do not stop
    beforehand!

**GOALS:**
Your primary objective is to optimize the 'solve' function to run as as
    fast as possible, while returning the optimal solution.

You will receive better scores the quicker your solution runs, and you
    will be penalized for exceeding the time limit or returning non-
    optimal solutions.

Below you find the description of the task you will have to solve. Read
    it carefully and understand what the problem is and what your solver
    should do.

<task/description.txt>

Here is the baseline which you will be graded against. Your task is to
    write a function that produces the same output, in less time.

<task.solve>

This function will be used to check if your solution is valid for a given
        problem. If it returns False, it means the solution is invalid:

<task.is_solution>

## B.2 AlgoTuner Interface Feedback

In this section, we show the type and style of feedback given back to the LM (within the running of AlgoTuner) in three different scenarios.

### B.2.1 Successful Edit and Evaluation

LM Message

```
I'll create our solver function in solver.py which transforms the generalized
    eigenvalue problem into a standard one using Cholesky decomposition and
    then solves it using NumPy's eigenvalue routines.
'''
edit
file: solver.py
lines: 1-100
---
import numpy as np
from typing import List, Tuple

class Solver:
    @staticmethod
    def solve(problem: Tuple[List[List[float]], List[List[float]]]) -> Tuple[
    List[float], List[List[float]]]:
        """
        Solve the generalized eigenvalue problem A x = lambda B x.

        Given:
          - A: a symmetric matrix,
          - B: a symmetric positive definite matrix,
        this function computes the eigenpairs (eigenvalues and eigenvectors)
    where the eigenvectors are
        B-normalized and mutually B-orthogonal.

        The eigenvalues are returned in descending order along with their
    corresponding eigenvectors.

        :param problem: Tuple (A, B), where A and B are given as list-of-
    lists.
        :return: tuple (eigenvalues_list, eigenvectors_list) where:
                - eigenvalues_list is a list of eigenvalues (sorted in
    descending order),
                - eigenvectors_list is a list of eigenvectors (each a list
    of floats) corresponding to those eigenvalues.
        """
        A, B = problem
        A = np.array(A, dtype=float)
        B = np.array(B, dtype=float)

        # Use Cholesky decomposition for numerical stability when dealing
    with B
        L = np.linalg.cholesky(B)
        Linv = np.linalg.inv(L)
        Atilde = Linv @ A @ Linv.T

        # Compute eigenvalues and eigenvectors
        eigenvalues, eigenvectors = np.linalg.eigh(Atilde)

        # Transform eigenvectors back to the original space
        eigenvectors = Linv.T @ eigenvectors

        # Normalize eigenvectors with respect to the B-inner product
        for i in range(eigenvectors.shape[1]):
```

```
            v = eigenvectors[:, i]
            norm = np.sqrt(np.dot(v, B @ v))
            if norm > 0:
                eigenvectors[:, i] = v / norm

        # Reverse to get descending order
        eigenvalues = eigenvalues[::-1]
        eigenvectors = eigenvectors[:, ::-1]

        # Convert to lists
        eigenvalues_list = eigenvalues.tolist()
        eigenvectors_list = [eigenvectors[:, i].tolist() for i in range(
    eigenvectors.shape[1])]

        return eigenvalues_list, eigenvectors_list
---
'''
```

**System Message**

You have sent 2 messages and have used up $0.0130. You have $0.9870 remaining
    .

Edit successful for solver.py.

Contents of solver.py (lines 1-50 out of 53)
(| = existing code, > = modified code)

```
> 01: import numpy as np
> 02: from typing import List, Tuple
> 03:
> 04: class Solver:
> 05:     @staticmethod
> 06:     def solve(problem: Tuple[List[List[float]], List[List[float]]]) ->
    Tuple[List[float], List[List[float]]]:
> 07:         """
> 08:         Solve the generalized eigenvalue problem A x = lambda B x.
> 09:
> 10:         Given:
> 11:             - A: a symmetric matrix,
> 12:             - B: a symmetric positive definite matrix,
> 13:         this function computes the eigenpairs (eigenvalues and
    eigenvectors) where the eigenvectors are
> 14:         B-normalized and mutually B-orthogonal.
> 15:
> 16:         The eigenvalues are returned in descending order along with
    their corresponding eigenvectors.
> 17:
> 18:         :param problem: Tuple (A, B), where A and B are given as list-
    of-lists.
> 19:         :return: tuple (eigenvalues_list, eigenvectors_list) where:
> 20:                     - eigenvalues_list is a list of eigenvalues (sorted in
     descending order),
> 21:                     - eigenvectors_list is a list of eigenvectors (each a
    list of floats) corresponding to those eigenvalues.
> 22:         """
> 23:         A, B = problem
> 24:         A = np.array(A, dtype=float)
> 25:         B = np.array(B, dtype=float)
> 26:
```

```
> 27:          # Use Cholesky decomposition for numerical stability when
     dealing with B
> 28:          L = np.linalg.cholesky(B)
> 29:          Linv = np.linalg.inv(L)
> 30:          Atilde = Linv @ A @ Linv.T
> 31:
> 32:          # Compute eigenvalues and eigenvectors
> 33:          eigenvalues, eigenvectors = np.linalg.eigh(Atilde)
> 34:
> 35:          # Transform eigenvectors back to the original space
> 36:          eigenvectors = Linv.T @ eigenvectors
> 37:
> 38:          # Normalize eigenvectors with respect to the B-inner product
> 39:          for i in range(eigenvectors.shape[1]):
> 40:              v = eigenvectors[:, i]
> 41:              norm = np.sqrt(np.dot(v, B @ v))
> 42:              if norm > 0:
> 43:                  eigenvectors[:, i] = v / norm
> 44:
> 45:          # Reverse to get descending order
> 46:          eigenvalues = eigenvalues[::-1]
> 47:          eigenvectors = eigenvectors[:, ::-1]
> 48:
> 49:          # Convert to lists
> 50:          eigenvalues_list = eigenvalues.tolist()
...

Speedup: 1.00x
  (Speedup = Baseline Time / Your Time; Higher is better)

  Valid Solutions: 100%
  Invalid Solutions: 0%
  Timeouts: 0%

Snapshot saved (Best speedup reached, state saved!)
```

### B.2.2   Linter Errors

**LM Message**

```
I'll create the solver.py file with our numba-accelerated implementation
    using the proper prepend command.

edit
file: solver.py
lines: 0-0
---
import numpy as np
from numba import njit

# Global up and down factors. These might be overridden externally.
up = 1
down = 1

@njit(cache=True)
def _upfirdn1d_numba(h, x, up, down):
    Lx = x.shape[0]
    Lh = h.shape[0]
    # The length of the convolved signal after upsampling.
```

```
        Lfull = (Lx - 1) * up + Lh
        # Compute the length of the output after downsampling.
        Lout = (Lfull - 1) // down + 1
        y = np.zeros(Lout, dtype=x.dtype)
        for n in range(Lout):
            pos = n * down
            s = 0.0
            kmin = 0
            if pos - (Lx - 1) * up > 0:
                kmin = pos - (Lx - 1) * up
            kmax = pos
            if kmax > Lh - 1:
                kmax = Lh - 1
            for k in range(kmin, kmax + 1):
                r = pos - k
                if r % up == 0:
                    m = r // up
                    if m < Lx:
                        s += x[m] * h[k]
            y[n] = s
        return y

def solve(problem: list) -> list:
    """
    Compute the upfirdn operation for each filter-signal pair in the problem
    list.

    Each tuple (h, x) is processed by upsampling x by factor 'up', filtering
    with h, and then downsampling by factor 'down'.

    :param problem: A list of tuples (h, x) where h and x are 1D arrays.
    :return: A list of 1D arrays representing the upfirdn results.
    """
    results = []
    for h, x in problem:
        h_arr = np.ascontiguousarray(np.array(h, dtype=np.float64))
        x_arr = np.ascontiguousarray(np.array(x, dtype=np.float64))
        res = _upfirdn1d_numba(h_arr, x_arr, up, down)
        results.append(res)
    return results
---
```

Edit failed (and thus not applied) for solver.py: Line 9: Redefining name 'up
' from outer scope (line 5) (redefined-outer-name)
Line 9: Redefining name 'down' from outer scope (line 6) (redefined-outer-
name)

PROPOSED CHANGES - This is what you tried to apply (lines 1-50 out of 50):
(| = existing code, > = modified code)

```
> 06:    down = 1
> 07:
> 08: @njit(cache=True)
> 09: def _upfirdn1d_numba(h, x, up, down):
> 10:     Lx = x.shape[0]
> 11:     Lh = h.shape[0]
> 12:     # The length of the convolved signal after upsampling.
> 13:     Lfull = (Lx - 1) * up + Lh
> 14:     # Compute the length of the output after downsampling.
> 15:     Lout = (Lfull - 1) // down + 1
> 16:     y = np.zeros(Lout, dtype=x.dtype)
> 17:     for n in range(Lout):
> 18:         pos = n * down
> 19:         s = 0.0
> 20:         kmin = 0
> 21:         if pos - (Lx - 1) * up > 0:
> 22:             kmin = pos - (Lx - 1) * up
> 23:         kmax = pos
> 24:         if kmax > Lh - 1:
> 25:             kmax = Lh - 1
> 26:         for k in range(kmin, kmax + 1):
> 27:             r = pos - k
> 28:             if r % up == 0:
> 29:                 m = r // up
> 30:                 if m < Lx:
> 31:                     s += x[m] * h[k]
> 32:         y[n] = s
> 33:     return y
> 34:
> 35: def solve(problem: list) -> list:
> 36:     """
> 37:     Compute the upfirdn operation for each filter-signal pair in the
     problem list.
> 38:
> 39:     Each tuple (h, x) is processed by upsampling x by factor 'up',
     filtering with h, and then downsampling by factor 'down'.
> 40:
> 41:     :param problem: A list of tuples (h, x) where h and x are 1D arrays
     .
> 42:     :return: A list of 1D arrays representing the upfirdn results.
> 43:     """
> 44:     results = []
> 45:     for h, x in problem:
> 46:         h_arr = np.ascontiguousarray(np.array(h, dtype=np.float64))
> 47:         x_arr = np.ascontiguousarray(np.array(x, dtype=np.float64))
> 48:         res = _upfirdn1d_numba(h_arr, x_arr, up, down)
> 49:         results.append(res)
> 50:     return results
```

Contents of current file:
File solver.py is empty.

### B.2.3 Runtime Errors

```
'''
eval_input [[1.2, -0.5], [0.3, 2.1]]
'''
```

```
You have sent 0 messages and have used up $0.2630. You have $0.7370 remaining
    .

Starting evaluation...

numpy.linalg.LinAlgError: 0-dimensional array given. Array must be at least
    two-dimensional in function 'solve' at line 22 in solver.py

Code Context:
   12:          """
   13:          Solve the eigenvector problem for the given non-symmetric
    matrix.
   14:          Compute eigenvalues and eigenvectors using np.linalg.eig.
   15:          Sort the eigenpairs in descending order by the real part (and
    then imaginary part) of the eigenvalues.
   16:          Return the eigenvectors (each normalized to unit norm) as a
    list of lists of complex numbers.
   17:
   18:          :param problem: A non-symmetric square matrix.
   19:          :return: A list of normalized eigenvectors sorted in
    descending order.
   20:          """
   21:          A = problem
 ! 22:          eigenvalues, eigenvectors = np.linalg.eig(A)
   23:          # Zip eigenvalues with corresponding eigenvectors (columns of
    eigenvectors matrix)
   24:          pairs = list(zip(eigenvalues, eigenvectors.T))
   25:          # Sort by descending order of eigenvalue real part, then
    imaginary part
   26:          pairs.sort(key=lambda pair: (-pair[0].real, -pair[0].imag))
   27:          sorted_evecs = []
   28:          for _, vec in pairs:
   29:              vec_arr = np.array(vec, dtype=complex)
   30:              norm = np.linalg.norm(vec_arr)
   31:              if norm > 1e-12:
   32:                  vec_arr = vec_arr / norm
```

# C AlgoTuner Trajectory Analysis

## C.1 Task Speedup Distribution

In Table 7 we report the distribution of speedups achieved by AlgoTuner when using each of the four models tested. We also include the *Task Best* result, which represents the performance obtained by selecting the best-performing model for each task.

Table 7: Outcome distribution per model, by highest achieved speedup. Columns report the percentage of submitted code speedups in each range, or N/A where the submitted code had errors or timeouts.

| Model | ≥1.1× (%) | 0.9×–1.1× (%) | <0.9× (%) | Invalid (%) |
|---|---|---|---|---|
| Claude Opus 4 | 39.6 | 31.8 | 3.2 | 25.3 |
| DeepSeek R1 | 61.0 | 26.0 | 6.5 | 6.5 |
| Gemini 2.5 Pro | 49.4 | 16.2 | 7.1 | 27.3 |
| o4-mini | 59.7 | 28.6 | 5.8 | 5.8 |
| **Overall** | **52.4** | **25.6** | **5.7** | **16.2** |
| **Task Best** | **74.7** | **19.5** | **3.9** | **1.9** |

## C.2 Development Set vs Test Set Performance

To assess the magnitude of overfitting on the development set of instances, we compare the speedups achieved by AlgoTuner's code on development and test instances for each model. For each task, we compute the offset as the ratio of development to test speedup minus one, (Dev Speedup/Test Speedup) − 1. Positive values indicate better performance on the development set, while negative values indicate better performance on the test set.

The small offsets shown in Table 8 indicate no meaningful overfitting.

Table 8: Offset values for each model. Positive values indicate higher speedups on developement instances; negative values indicate higher speedups on test instances.

| Model Name | Median Offset | Mean Offset |
|---|---|---|
| DeepSeek R1 | +0.016 | -0.131 |
| o4-mini | +0.005 | -1.385 |
| Claude Opus 4 | +0.000 | -0.507 |
| Gemini 2.5 Pro | -0.021 | -0.043 |

## C.3 AlgoTuner Trajectory Patterns

In Table 9, we show how speedups change over time as AlgoTuner runs. We compare the first evaluation it performs to its best evaluation and classify the results into the following categories:

- **Significant:** Speedup of $1.1\times$ or greater
- **Insignificant:** Speedup between $0.9\times$ and $1.1\times$
- **Slow:** Speedup less than $0.9\times$
- **Invalid:** No valid speedup measurement (at least one timeout or invalid result)

where speedup is relative to the reference solver.

Table 9: AlgoTuner's first and best evaluations for each model, showing how tasks transition between the four speedup categories: Significant, Insignificant, Slow, and Invalid.

| Model | Sig. From Beginning | Invalid→Sig. | Insig.→Sig. | Slow→Sig. | Always Insig. | Invalid→Insig. | Never Had Success |
|-------|------|------|------|------|------|------|------|
| Claude Opus 4 | 16.2 | 7.8 | 16.9 | 2.0 | 31.8 | 5.2 | 20.1 |
| DeepSeek R1 | 27.9 | 18.8 | 14.9 | 7.2 | 11.1 | 11.0 | 9.1 |
| Gemini 2.5 Pro | 36.4 | 9.1 | 9.7 | 2.6 | 12.3 | 4.6 | 25.3 |
| o4-mini | 33.1 | 13.6 | 13.6 | 7.2 | 10.4 | 14.3 | 7.8 |
| **Overall** | **28.4** | **12.3** | **13.8** | **4.7** | **16.4** | **8.8** | **15.6** |

In Table 10, we show how AlgoTuner's package usage evolves over time. For each model, we consider tasks where the best speedup reached at least $1.1\times$, listing the packages used in the first evaluation to reach that threshold and in the evaluation with the maximum speedup, along with the reference packages for those tasks.

Table 10: Per-package outcomes for each model. For each package, we compare the first and best evaluations achieved by AlgoTuner, with outcomes classified as Significant, Insignificant, Slow, or Invalid.

(a) Claude Opus 4

| Package | Reference | First ≥ 1.1× | Max Speedup | Δ (Max-First) |
|---------|-----------|--------------|-------------|---------------|
| jax_jit | 154 | 64 | 149 | +85 |
| numpy | 132 | 57 | 130 | +73 |
| scipy | 61 | 31 | 69 | +38 |
| cvxpy | 27 | 5 | 17 | +12 |
| numba | 1 | 15 | 25 | +10 |
| numba_jit | 1 | 8 | 15 | +7 |
| ortools | 14 | 7 | 11 | +4 |
| vectorization | 3 | 6 | 10 | +4 |
| networkx | 12 | 1 | 4 | +3 |
| jax | 0 | 3 | 4 | +1 |

(b) DeepSeek R1

| Package | Reference | First ≥1.1× | Max Speedup | Δ (Max-First) |
|---------|-----------|-------------|-------------|---------------|
| jax_jit | 154 | 91 | 147 | +56 |
| numpy | 132 | 77 | 122 | +45 |
| scipy | 61 | 34 | 63 | +29 |
| cvxpy | 27 | 6 | 15 | +9 |
| numba | 1 | 30 | 37 | +7 |
| numba_jit | 1 | 28 | 35 | +7 |
| vectorization | 3 | 13 | 18 | +5 |
| jax | 0 | 8 | 10 | +2 |
| cryptography | 3 | 0 | 2 | +2 |
| sklearn | 9 | 5 | 6 | +1 |

(c) Gemini 2.5 Pro

| Package | Reference | First ≥1.1× | Max Speedup | Δ (Max-First) |
|---------|-----------|-------------|-------------|---------------|
| jax_jit | 154 | 84 | 154 | +70 |
| numpy | 132 | 67 | 129 | +62 |
| scipy | 61 | 39 | 78 | +39 |
| numba | 1 | 21 | 31 | +10 |
| numba_jit | 1 | 20 | 30 | +10 |
| vectorization | 3 | 15 | 19 | +4 |
| ortools | 14 | 11 | 14 | +3 |
| jax | 0 | 2 | 5 | +3 |
| sklearn | 9 | 0 | 3 | +3 |
| cvxpy | 27 | 5 | 7 | +2 |

(d) o4-mini

| Package | Reference | First ≥1.1× | Max Speedup | Δ (Max-First) |
|---------|-----------|-------------|-------------|---------------|
| jax_jit | 154 | 97 | 154 | +57 |
| numpy | 132 | 73 | 118 | +45 |
| scipy | 61 | 39 | 67 | +28 |
| cvxpy | 27 | 3 | 10 | +7 |
| numba | 1 | 19 | 25 | +6 |
| numba_jit | 1 | 17 | 22 | +5 |
| vectorization | 3 | 5 | 8 | +3 |
| ortools | 14 | 10 | 12 | +2 |
| sklearn | 9 | 3 | 5 | +2 |
| threading | 0 | 1 | 3 | +2 |

# D Python Packages

In Table 11 we show the Python packages used AlgoTune, as well as packages installed on the AlgoTuner agent interface.

Table 11: Python packages used in the AlgoTune benchmark, installed on the AlgoTune Agent interface, and their open-source licenses.

| Package | AlgoTune (Benchmark) | AlgoTuner (Agent) | License |
|---|:---:|:---:|:---:|
| NumPy (Harris et al., 2020) | ✓ | ✓ | BSD 3-Clause |
| SciPy (Virtanen et al., 2020) | ✓ | ✓ | BSD 3-Clause |
| Pandas (McKinney, 2010) | ✗ | ✓ | BSD 3-Clause |
| Cython (Behnel et al., 2011) | ✗ | ✓ | Apache 2.0 |
| Numba (Lam et al., 2015) | ✗ | ✓ | BSD 2-Clause |
| Dask (Rocklin, 2015) | ✗ | ✓ | BSD 3-Clause |
| PuLP (Mitchell et al., 2009) | ✓ | ✓ | MIT |
| OR-Tools (Google, 2020) | ✓ | ✓ | Apache 2.0 |
| Pyomo (Hart et al., 2011) | ✗ | ✓ | BSD 3-Clause |
| HiGHS / HighSpy (Huangfu and Hall, 2018) | ✗ | ✓ | MIT |
| NetworkX (Hagberg et al., 2008) | ✓ | ✓ | BSD 3-Clause |
| python-sat (Biere et al., 2012) | ✓ | ✓ | MIT |
| JAX (Bradbury et al., 2018) | ✗ | ✓ | Apache 2.0 |
| Diffrax (Kidger, 2021) | ✓ | ✓ | Apache 2.0 |
| CVXPY (Agrawal et al., 2018; Diamond and Boyd, 2016) | ✓ | ✓ | Apache 2.0 |
| Pythran (Guelton et al., 2015) | ✓ | ✓ | BSD 3-Clause |
| Dace (Ben-Nun et al., 2019) | ✓ | ✓ | BSD 3-Clause |

# E    Performance Improvements in Python Repositories

In this section, we show a sample of performance improving pull requests to three Python repositories: NumPy, SciPy, and NetworkX. All of the below pull requests were submitted in the past two years, and greatly increase performance (as reported in the PR itself). For each PR, we highlight the most significant reported performance greatest improvement.

NumPy:

- **ENH: Add a fast-path for `ufunc.at` on aligned 1D arrays** (PR #22889): Up to 6.3x faster when no casting is needed on 1D aligned inputs (e.g. bench_ufunc.At.time_sum_at dropped from $54.0 \pm 0.2$ ms to $8.42 \pm 0.02$ ms). https://github.com/numpy/numpy/pull/22889

- **ENH: Vectorize quicksort for 16-bit and 64-bit dtype using AVX512** (PR #22315): Up to 15x speedup for 16-bit sorts and 9x speedup for 64-bit sorts on AVX-512-capable CPUs. https://github.com/numpy/numpy/pull/22315

- **ENH: Accelerate `unique` for integer dtypes via hash tables** (PR #26018): Roughly 2.7x speedup on 1 billion random integers (unique count in 7.815 s vs. 21.436 s for the previous implementation). https://github.com/numpy/numpy/pull/26018

SciPy:

- **ENH: Vectorize `stats.mannwhitneyu`** (PR #19749): Vectorizes the statistic calculation, achieving up to ~21x speedup (1.38 s → 64.4 ms in certain cases). https://github.com/scipy/scipy/pull/19749

- **ENH: Vectorize `stats.rankdata`** (PR #19776): Vectorizes rankdata along an axis, yielding up to ~ 296x faster runtimes (2.58 ms → 8.7 $\mu$s for a $(100, 100)$ array). https://github.com/scipy/scipy/pull/19776

- **ENH: Fast-path for sparse Frobenius norm** (PR #14317): Directly accesses the data array to compute the norm, resulting in up to 5x speedup in some cases. https://github.com/scipy/scipy/pull/14317

NetworkX:

- **BUG: Fix `weakly_connected_components()` performance on graph views** (PR #7586): Moves the repeated len(G) call outside the loop, cutting runtime from $\sim 15.4$ s to $0.064$ s per iteration, over $240\times$ faster. https://github.com/networkx/networkx/pull/7586

- **ENH: Speed up `harmonic_centrality`** (PR #7595): Implements graph reversal for node-subset queries, reducing computation on large wheel graphs from 95.9ms to 717$\mu$s, 134x faster. https://github.com/networkx/networkx/pull/7595

- **ENH: Speed up `common_neighbors` / `non_neighbors`** (PR #7244): Replaces generator-based neighbor lookups with direct _adj dict operations, achieving up to ~600x speedup on star-center queries and around 11x on complete-graph common neighbors. https://github.com/networkx/networkx/pull/7244

# F AlgoTune vs KernelBench

Concurrent work, KernelBench (Ouyang et al., 2025) is similar to AlgoTune: both are code optimization benchmarks. In this section, we summarize the main differences between them.

KernelBench is made up of 250 GPU kernels, where the goal is to write highly optimized low level code that speeds up their runtime, while still producing correct outputs. KernelBench is split into three levels based on kernel complexity. The first level contains simple funcitons like `softmax` or `tanh`, while the third level contains more complex kernels like `ShuffleNet` or `LSTM`.

This approach has two downsides: first, the runtimes of kernels in the benchmark is highly varied; level 1 kernels run in microseconds, while level 3 kernels run in milliseconds (see Table 12). 40.8% of the kernels in KernelBench run in under 0.1 milliseconds, while the rest take between 0.1 and 100 milliseconds to run. Kernels with low runtimes are harder to optimize, as the process overhead takes a significant part of the runtime. This makes the comparison between the improvement of different kernel runtimes somewhat complicated.

In contrast, AlgoTune's tasks have controllable runtimes, which results in the benchmark having more uniform runtimes (see H). Importantly, AlgoTune covers a broage range of functions in math, science, computer science, machine learning, and more (see §1 for a discussion).

Table 12: Number of kernels per time interval as reported by Ouyang (2025), for KernelBench (Ouyang et al., 2025), by level.

| Time Interval | Level 1 (100 ops) | Level 2 (100 ops) | Level 3 (50 ops) | Pct of Total [%] |
|---|---|---|---|---|
| 10–20 $\mu$s | 21 | 0 | 0 | 8.4 |
| 20–50 $\mu$s | 24 | 12 | 0 | 14.4 |
| 50–100 $\mu$s | 4 | 37 | 4 | 18.0 |
| 0.1–1 ms | 22 | 21 | 8 | 20.4 |
| 1–10 ms | 23 | 20 | 20 | 25.2 |
| 10–100 ms | 6 | 10 | 18 | 13.6 |

# G Task Size Determination

Algorithm 1 shows the two phase search algorithm used to find the problem size parameter $n$ for each Task in the benchmark.

```python
def find_n_for_time(task, target_time,
                    n_min=1, n_max=10**7,
                    log_sweep=16, refine=8,
                    m=10, seed=1, runs=5, warmups=3,
                    mem_mb=8_192):
    cache = {}
    def probe(n):
        if n not in cache:
            mean, stats = measure_solve_time(
                task, n, target_time, m, seed,
                timing_num_runs=runs,
                timing_warmup_runs=warmups,
                timeout_s=max(1, 50 * target_time),
                memory_limit_mb=mem_mb,
            )
            cache[n] = (mean, stats)
        return cache[n]

    grid = sorted({n_min, *map(int,
                        np.geomspace(n_min, n_max, log_sweep)), n_max})
    best = (None, float('inf'))
    low_ok = high_fail = None

    for n in grid:
        mean, _ = probe(n)
        if mean is None or mean > target_time:
            if low_ok is not None:
                high_fail = n
                break
            continue
        low_ok = n
        err = abs(mean - target_time)
        best = min(best, (n, err), key=lambda p: p[1])

    if best[0] is None:
        return None

    for _ in range(refine):
        if high_fail is None or high_fail - low_ok <= 1:
            break
        mid = (low_ok + high_fail) // 2
        mean, _ = probe(mid)
        if mean is None:
            high_fail = mid - 1
            continue
        err = abs(mean - target_time)
        best = min(best, (mid, err), key=lambda p: p[1])
        if mean > target_time:
            high_fail = mid - 1
        else:
            low_ok = mid

    return best[0]
```

Algorithm 1: Python pseudocode for selecting the $n$ parameter value whose average `solve()` runtime is closest to the target time, for a given task.

# H Task Timings

We report the size parameter $n$ and per task timings in Table 13. The average time per task is calculated by averaging three runs, in which the average time over the 100 development instances is calculated.

Table 13: Per task size parameter $n$ values and average time for the reference solve function, across three timing runs.

| Task | n | Average Time (ms) |
|------|---|-------------------|
| aes_gcm_encryption | 291598 | 203.78 ± 2.72 |
| affine_transform_2d | 1123 | 111.78 ± 0.12 |
| aircraft_wing_design | 10 | 99.05 ± 1.04 |
| articulation_points | 837 | 102.30 ± 0.54 |
| base64_encoding | 48512 | 142.39 ± 0.18 |
| battery_scheduling | 6 | 108.82 ± 1.21 |
| btsp | 14 | 13.55 ± 0.01 |
| capacitated_facility_location | 4 | 81.20 ± 0.41 |
| chacha_encryption | 197380 | 209.56 ± 0.57 |
| channel_capacity | 162 | 102.53 ± 0.10 |
| chebyshev_center | 206 | 98.16 ± 0.18 |
| cholesky_factorization | 1660 | 109.50 ± 0.22 |
| clustering_outliers | 2457 | 98.79 ± 0.02 |
| communicability | 61 | 97.93 ± 0.46 |
| convex_hull | 267021 | 29.86 ± 0.08 |
| convolve2d_full_fill | 6 | 148.32 ± 0.85 |
| convolve_1d | 72989 | 146.07 ± 20.58 |
| correlate2d_full_fill | 6 | 139.51 ± 0.07 |
| correlate_1d | 1504 | 119.68 ± 0.59 |
| count_connected_components | 1707 | 90.68 ± 2.27 |
| count_riemann_zeta_zeros | 15849 | 71.92 ± 0.07 |
| cumulative_simpson_1d | 4443523 | 93.56 ± 0.94 |
| cumulative_simpson_multid | 423 | 105.07 ± 0.91 |
| cvar_projection | 9 | 87.69 ± 0.05 |
| cyclic_independent_set | 4 | 91.39 ± 0.60 |
| dct_type_I_scipy_fftpack | 1958 | 138.96 ± 0.49 |
| delaunay | 21339 | 264.79 ± 0.71 |
| dijkstra_from_indices | 5271 | 103.11 ± 0.37 |
| discrete_log | 25 | 9.83 ± 0.10 |
| dst_type_II_scipy_fftpack | 2054 | 88.09 ± 0.42 |
| dynamic_assortment_planning | 28 | 68.60 ± 0.14 |
| earth_movers_distance | 1151 | 103.48 ± 2.17 |
| edge_expansion | 4408 | 36.32 ± 0.19 |
| eigenvalues_complex | 474 | 99.81 ± 0.44 |
| eigenvalues_real | 875 | 124.97 ± 0.91 |
| eigenvectors_complex | 463 | 101.07 ± 0.36 |
| eigenvectors_real | 827 | 110.33 ± 8.47 |
| elementwise_integration | 372 | 100.14 ± 0.11 |
| feedback_controller_design | 15 | 115.87 ± 0.13 |
| fft_cmplx_scipy_fftpack | 1860 | 82.86 ± 2.46 |
| fft_convolution | 542069 | 107.89 ± 1.55 |
| fft_real_scipy_fftpack | 2738 | 136.69 ± 0.45 |
| firls | 1113 | 103.26 ± 0.67 |
| generalized_eigenvalues_complex | 272 | 99.63 ± 1.14 |
| generalized_eigenvalues_real | 668 | 102.24 ± 0.40 |
| generalized_eigenvectors_complex | 269 | 99.38 ± 1.09 |
| generalized_eigenvectors_real | 574 | 107.08 ± 0.86 |
| graph_coloring_assign | 38 | 76.27 ± 0.33 |
| graph_global_efficiency | 507 | 101.76 ± 0.35 |
| graph_isomorphism | 131 | 99.58 ± 0.35 |
| graph_laplacian | 44505 | 101.48 ± 0.20 |

| Task | $n$ | Average Time (ms) |
|------|-----|-------------------|
| group_lasso | 144 | $101.06 \pm 0.12$ |
| gzip_compression | 658 | $100.41 \pm 0.05$ |
| integer_factorization | 132 | $54.82 \pm 0.02$ |
| job_shop_scheduling | 15 | $64.12 \pm 0.49$ |
| kalman_filter | 23 | $100.16 \pm 0.94$ |
| kcenters | 49 | $84.44 \pm 0.12$ |
| kd_tree | 198 | $112.55 \pm 0.51$ |
| kernel_density_estimation | 300 | $42.69 \pm 0.05$ |
| kmeans | 278 | $114.20 \pm 0.84$ |
| ks_test_2samp | 359188 | $86.17 \pm 0.06$ |
| l0_pruning | 695029 | $102.98 \pm 0.47$ |
| l1_pruning | 473085 | $107.67 \pm 0.29$ |
| lasso | 398 | $120.11 \pm 1.88$ |
| least_squares | 102713 | $111.89 \pm 0.19$ |
| linear_system_solver | 1450 | $103.18 \pm 1.65$ |
| lp_box | 210 | $103.06 \pm 1.18$ |
| lp_centering | 215 | $98.92 \pm 0.18$ |
| lp_mdp | 10 | $92.34 \pm 1.90$ |
| lqr | 111 | $102.41 \pm 0.30$ |
| lti_simulation | 24921 | $193.19 \pm 0.50$ |
| lu_factorization | 1104 | $113.38 \pm 2.43$ |
| lyapunov_stability | 17 | $40.50 \pm 0.16$ |
| markowitz | 382 | $94.02 \pm 0.16$ |
| matrix_completion | 15 | $92.05 \pm 0.25$ |
| matrix_exponential | 555 | $106.60 \pm 1.27$ |
| matrix_exponential_sparse | 318 | $105.02 \pm 0.20$ |
| matrix_multiplication | 790 | $105.85 \pm 0.14$ |
| matrix_sqrt | 281 | $101.31 \pm 0.22$ |
| max_clique_cpsat | 12 | $35.99 \pm 0.22$ |
| max_common_subgraph | 4 | $28.50 \pm 0.27$ |
| max_flow_min_cost | 64 | $107.77 \pm 0.35$ |
| max_independent_set_cpsat | 12 | $24.95 \pm 0.06$ |
| max_weighted_independent_set | 61 | $35.35 \pm 0.62$ |
| min_dominating_set | 9 | $27.46 \pm 0.07$ |
| min_weight_assignment | 756 | $97.28 \pm 0.35$ |
| minimum_spanning_tree | 571 | $275.07 \pm 0.89$ |
| minimum_volume_ellipsoid | 28 | $99.64 \pm 2.63$ |
| multi_dim_knapsack | 25 | $18.43 \pm 0.24$ |
| nmf | 7 | $104.67 \pm 0.05$ |
| ode_brusselator | 199 | $102.15 \pm 0.61$ |
| ode_fitzhughnagumo | 15 | $87.32 \pm 1.34$ |
| ode_hires | 370 | $109.68 \pm 0.98$ |
| ode_hodgkinhuxley | 43 | $96.39 \pm 0.39$ |
| ode_lorenz96_nonchaotic | 7856 | $102.04 \pm 0.51$ |
| ode_lotkavolterra | 161 | $97.44 \pm 0.33$ |
| ode_nbodyproblem | 8 | $98.31 \pm 0.13$ |
| ode_seirs | 1971 | $102.58 \pm 1.29$ |
| ode_stiff_robertson | 9999999 | $89.59 \pm 0.18$ |
| ode_stiff_vanderpol | 2 | $119.64 \pm 1.06$ |
| odr | 31132 | $60.32 \pm 0.04$ |
| optimal_advertising | 43 | $114.93 \pm 0.36$ |
| outer_product | 10630 | $106.10 \pm 5.72$ |
| pagerank | 4798 | $49.68 \pm 0.16$ |
| pca | 34 | $91.12 \pm 0.93$ |
| pde_burgers1d | 12 | $116.76 \pm 0.87$ |
| pde_heat1d | 8 | $88.76 \pm 0.81$ |
| polynomial_mixed | 415 | $103.97 \pm 1.15$ |
| polynomial_real | 396 | $99.36 \pm 0.03$ |
| power_control | 98 | $102.40 \pm 0.34$ |
| procrustes | 585 | $101.25 \pm 0.56$ |
| psd_cone_projection | 349 | $101.89 \pm 0.23$ |

Table 13 – continued from previous page

| Task | $n$ | Average Time (ms) |
|---|---|---|
| qp | 278 | 99.31 ± 0.14 |
| qr_factorization | 971 | 105.48 ± 0.83 |
| quantile_regression | 356 | 101.68 ± 0.23 |
| queens_with_obstacles | 9 | 27.01 ± 0.12 |
| queuing | 665036 | 104.97 ± 0.58 |
| qz_factorization | 272 | 98.93 ± 0.11 |
| randomized_svd | 776 | 105.83 ± 0.10 |
| rbf_interpolation | 68 | 38.94 ± 0.02 |
| rectanglepacking | 8 | 25.55 ± 0.08 |
| robust_kalman_filter | 15 | 94.21 ± 0.51 |
| robust_linear_program | 12 | 104.93 ± 0.95 |
| rocket_landing_optimization | 102 | 97.94 ± 0.38 |
| rotate_2d | 1086 | 108.52 ± 0.28 |
| set_cover | 52 | 70.74 ± 0.04 |
| set_cover_conflicts | 36 | 26.38 ± 0.12 |
| sha256_hashing | 183042 | 170.97 ± 0.67 |
| shift_2d | 1047 | 83.53 ± 0.98 |
| shortest_path_dijkstra | 352 | 100.16 ± 0.25 |
| sinkhorn | 1813 | 38.67 ± 1.88 |
| sparse_eigenvectors_complex | 1294 | 80.62 ± 0.10 |
| sparse_lowest_eigenvalues_posdef | 1341 | 111.01 ± 0.24 |
| sparse_lowest_eigenvectors_posdef | 1341 | 98.81 ± 0.14 |
| sparse_pca | 662 | 107.30 ± 0.15 |
| spectral_clustering | 8 | 57.19 ± 0.12 |
| stable_matching | 1209 | 102.95 ± 2.48 |
| svd | 474 | 117.63 ± 0.48 |
| svm | 571 | 82.83 ± 0.16 |
| sylvester_solver | 207 | 99.81 ± 0.26 |
| tensor_completion_3d | 6 | 140.06 ± 0.49 |
| toeplitz_solver | 8588 | 100.04 ± 0.03 |
| tsp | 27 | 50.35 ± 0.78 |
| two_eigenvalues_around_0 | 1123 | 100.67 ± 0.20 |
| unit_simplex_projection | 982958 | 104.50 ± 0.29 |
| upfirdn1d | 2582 | 116.45 ± 0.37 |
| vector_quantization | 166 | 104.60 ± 0.05 |
| vectorized_newton | 710026 | 103.55 ± 1.93 |
| vehicle_routing | 9 | 62.94 ± 1.68 |
| vertex_cover | 15 | 94.77 ± 0.28 |
| voronoi_diagram | 8997 | 129.64 ± 0.73 |
| wasserstein_dist | 64597 | 85.21 ± 0.04 |
| water_filling | 3865 | 102.03 ± 0.23 |
| zoom_2d | 971 | 104.78 ± 1.01 |

