# A   Results

In Table 4, we show a summary of the results of AlgoTuner for each of the four frontier models tests.
In Table 5, we detail the per-task timings for every model and task.

Table 4: AlgoTuner speedup when using each LM, with a budget of $0.5 for each task. Speedup percentage is calculated as the percentage of tasks for which AlgoTuner gets at least a $1.1\times$ speedup.

|  | o4-mini-high | R1 | Claude 3.7 Sonnet | Gemini 2.5 Pro |
|---|---|---|---|---|
| Pct. of Tasks Sped Up | 53.3% | 42.5% | 34.2% | 34.2% |

Table 5: Per task speedup for AlgoTuner, using four frontier LMs. Speedup is calculated as the ratio between the reference solve function's time and the LM-generated solve function's time.

| Task | o4-mini-high | R1 | Claude 3.7 Sonnet | Gemini 2.5 Pro |
|---|---|---|---|---|
| affine_transform_2d | 1.00 | 1.00 | 1.00 | 1.00 |
| articulation_points | 4.75 | 1.00 | 1.45 | 1.00 |
| base64_encoding | 1.01 | 1.00 | 1.00 | 1.00 |
| btsp | 74.47 | 4.02 | 3.23 | 6.74 |
| capacitated_facility_location | 8.16 | 8.19 | 1.00 | 1.00 |
| chacha_encryption | 1.00 | 1.00 | 1.00 | 1.00 |
| channel_capacity | 1.00 | 1.73 | 1.71 | 1.57 |
| chebyshev_center | 1.78 | 2.21 | 1.23 | 1.13 |
| cholesky_factorization | 2.17 | 1.00 | 1.00 | 1.00 |
| clustering_outliers | 1.11 | 1.00 | 1.06 | 1.18 |
| communicability | 217.10 | 162.59 | 183.45 | 166.40 |
| convex_hull | 2.18 | 1.00 | 1.00 | 1.00 |
| convex_quadratic_check | 96.58 | 2.52 | 6.05 | 6.62 |
| convolve2d_full_fill | 272.12 | 229.80 | 227.84 | 229.30 |
| convolve_1d | 1.00 | 1.00 | 1.00 | 1.00 |
| correlate_1d | 1.62 | 1.00 | 1.54 | 1.00 |
| count_connected_components | 6.86 | 1.00 | 3.92 | 3.44 |
| count_riemann_zeta_zeros | 1.00 | 1.00 | 1.02 | 1.00 |
| crew_pairing | 1.00 | 1.00 | 1.00 | 1.00 |
| cumulative_simpson_multid | 1.00 | 1.00 | 1.00 | 1.00 |
| cyclic_independent_set | 1.00 | 1.00 | 1.00 | 1.00 |
| determinant_matrix_exponential | 416.78 | 1.00 | 277.06 | 395.17 |
| dijkstra_from_indices | 1.64 | 1.00 | 1.00 | 1.00 |
| discrete_log | 1.28 | 1.00 | 1.00 | 1.00 |
| dynamic_assortment_planning | 2.45 | 1.08 | 1.00 | 28.82 |
| earth_movers_distance | 1.00 | 1.17 | 1.09 | 1.00 |
| edge_expansion | 25.22 | 1.00 | 1.00 | 1.00 |
| efficiency | 71.63 | 2.76 | 1.07 | 1.11 |
| eigenvalues_complex | 1.47 | 1.52 | 1.47 | 1.45 |
| eigenvalues_real | 2.43 | 2.44 | 2.39 | 2.41 |
| eigenvectors_complex | 1.03 | 1.00 | 1.00 | 1.02 |
| eigenvectors_real | 1.03 | 1.06 | 1.01 | 1.02 |
| elementwise_integration | 1.01 | 1.00 | 1.00 | 1.00 |
| feedback_controller_design | 119.25 | 1.00 | 1.00 | 1.00 |
| fft_real_scipy_fftpack | 2.98 | 2.66 | 1.00 | 1.00 |
| generalized_eigenvalues_complex | 3.84 | 2.04 | 2.04 | 2.02 |
| generalized_eigenvalues_real | 2.55 | 2.98 | 2.52 | 1.00 |
| generalized_eigenvectors_complex | 2.84 | 1.06 | 1.06 | 1.05 |
| generalized_eigenvectors_real | 1.72 | 1.71 | 1.52 | 1.15 |
| graph_coloring | 27.19 | 1.18 | 1.00 | 1.08 |
| graph_isomorphism | 56.73 | 19.68 | 24.62 | 1.98 |
| graph_laplacian | 1.48 | 1.00 | 1.08 | 1.57 |

Continued on next page

Table 5 – continued from previous page

| Task | o4-mini-high | R1 | Claude 3.7 Sonnet | Gemini 2.5 Pro |
| --- | --- | --- | --- | --- |
| gzip_compression | 1.00 | 1.00 | 1.00 | 1.00 |
| job_shop_scheduling | 1.42 | 1.75 | 1.40 | 1.48 |
| kalman_filter | 71.08 | 2.96 | 1.00 | 1.00 |
| kcenters | 1.00 | 1.00 | 1.00 | 1.00 |
| kd_tree | 1.23 | 1.20 | 1.00 | 1.01 |
| kernel_density_estimation | 1.00 | 1.00 | 1.00 | 1.00 |
| kmeans | 32.30 | 1.00 | 1.00 | 1.02 |
| ks_test_2samp | 1.00 | 1.03 | 1.03 | 1.00 |
| l0_pruning | 3.98 | 3.15 | 1.00 | 2.68 |
| l1_pruning | 2.94 | 2.84 | 1.27 | 1.40 |
| lasso | 1.00 | 1.00 | 1.00 | 1.00 |
| linear_system_solver | 1.10 | 1.12 | 1.09 | 1.09 |
| lp_box | 2.26 | 14.19 | 1.53 | 2.25 |
| lp_centering | 1.00 | 1.00 | 1.00 | 1.00 |
| lqr | 1.05 | 1.00 | 1.10 | 1.00 |
| lu_factorization | 1.00 | 1.00 | 1.00 | 1.00 |
| markowitz | 1.00 | 1.00 | 1.00 | 1.01 |
| matrix_exponential | 1.01 | 1.00 | 3.79 | 1.00 |
| matrix_multiplication | 1.08 | 1.06 | 1.09 | 1.08 |
| matrix_sqrt | 1.02 | 1.00 | 1.04 | 1.00 |
| max_clique | 21.15 | 1.26 | 3.80 | 2.66 |
| max_common_subgraph | 20.94 | 1.00 | 1.56 | 1.47 |
| max_flow_min_cost | 3.04 | 20.46 | 2.09 | 1.00 |
| max_independent_set | 1.00 | 1.63 | 1.00 | 1.51 |
| max_weighted_independent_set | 71.37 | 9.55 | 1.04 | 1.00 |
| min_weight_assignment | 1.42 | 1.25 | 1.00 | 1.00 |
| minimum_spanning_tree | 30.88 | 3.94 | 33.52 | 12.50 |
| minimum_volume_ellipsoid | 1.00 | 1.00 | 1.00 | 1.00 |
| multi_dim_knapsack | 1.52 | 1.48 | 1.37 | 1.00 |
| nmf | 1.00 | 1.00 | 1.00 | 1.00 |
| ode_fitzhughnagumo | 1.00 | 1.00 | 1.00 | 1.00 |
| ode_hires | 10.72 | 5.58 | 1.00 | 5.49 |
| ode_lorenz96_nonchaotic | 1.00 | 1.00 | 1.00 | 1.00 |
| ode_nbodyproblem | 1.00 | 1.00 | 1.00 | 1.00 |
| ode_stiff_robertson | 18.79 | 1.00 | 1.66 | 4.89 |
| ode_stiff_vanderpol | 1.00 | 1.00 | 1.00 | 1.00 |
| odr | 1.01 | 1.00 | 1.00 | 1.00 |
| pagerank | 1.00 | 1.00 | 1.00 | 1.57 |
| pca | 3.77 | 1.00 | 3.60 | 1.54 |
| pde_burgers1d | 1.00 | 1.00 | 1.00 | 1.00 |
| pde_heat1d | 1.00 | 1.00 | 1.00 | 1.00 |
| polynomial_mixed | 1.02 | 1.01 | 1.00 | 1.01 |
| polynomial_real | 23.60 | 142.29 | 1.00 | 1.00 |
| portfolio_optimization_cvar | 1.00 | 10.72 | 8.25 | 1.15 |
| procrustes | 1.23 | 1.04 | 1.03 | 1.04 |
| psd_cone_projection | 10.16 | 8.78 | 9.30 | 7.34 |
| qp | 1.75 | 1.76 | 1.01 | 1.00 |
| qr_factorization | 1.98 | 1.17 | 1.16 | 1.00 |
| quantile_regression | 1.27 | 1.00 | 1.00 | 1.03 |
| queens_with_obstacles | 1.42 | 2.40 | 1.00 | 1.38 |
| queuing | 76.82 | 6.72 | 1.00 | 1.00 |
| qz_factorization | 1.00 | 1.01 | 1.01 | 1.01 |
| randomized_svd | 5.66 | 3.84 | 1.26 | 1.98 |
| rbf_interpolation | 1.00 | 1.00 | 1.00 | 1.00 |
| rectanglepacking | 1.67 | 7.54 | 1.00 | 1.55 |
| robust_linear_program | 1.00 | 1.07 | 458.28 | 1.00 |
| rotate_2d | 1.00 | 1.00 | 1.00 | 1.00 |
| set_cover_conflicts | 31.03 | 4.97 | 4.49 | 4.92 |
| set_cover | 1.84 | 1.35 | 1.44 | 1.34 |
| sha256_hashing | 1.00 | 1.00 | 1.00 | 1.00 |
| shift_2d | 1.00 | 1.00 | 1.00 | 1.00 |

| Task | o4-mini-high | R1 | Claude 3.7 Sonnet | Gemini 2.5 Pro |
|---|---|---|---|---|
| shortest_path_dijkstra | 2.31 | 2.50 | 53.17 | 2.46 |
| sparse_eigenvectors_complex | 1.00 | 1.00 | 1.00 | 1.00 |
| spectral_clustering | 1.00 | 68.34 | 1.00 | 1.00 |
| stable_matching | 1.05 | 1.00 | 1.00 | 1.00 |
| toeplitz_solver | 1.00 | 1.00 | 1.00 | 1.00 |
| tsp | 1.00 | 1.27 | 1.00 | 1.09 |
| two_eigenvalues_around_0 | 1.41 | 1.65 | 1.01 | 1.58 |
| unit_simplex_projection | 1.92 | 1.05 | 1.05 | 1.05 |
| vector_quantization | 1.00 | 1.00 | 1.00 | 1.00 |
| vectorized_newton | 1.00 | 1.00 | 1.00 | 1.00 |
| vehicle_routing | 1.00 | 1.17 | 1.00 | 1.00 |
| vehicle_routing_circuit | 1.00 | 1.47 | 1.01 | 1.00 |
| vertex_cover | 12.04 | 1.07 | 12.52 | 1.23 |
| vertex_cover_cpsat | 14.36 | 1.44 | 1.00 | 1.19 |
| wasserstein_dist | 8.15 | 7.52 | 1.00 | 6.78 |
| water_filling | 418.00 | 346.88 | 109.62 | 81.21 |
| zoom_2d | 1.00 | 1.00 | 1.00 | 1.00 |

886

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

    ---
```

1124

## System Message

Edit failed (and thus not applied) for solver.py: Line 9: Redefining name 'up'
    from outer scope (line 5) (redefined-outer-name)
Line 9: Redefining name 'down' from outer scope (line 6) (redefined-outer-name)

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

---

**Algorithm 1** FINDKFORTIME — choose $n$ whose mean solve time best matches target time $\tau$

---

**Require:** task $T$; target time $\tau$; bounds $[k_{\min}, k_{\max}]$; examples $m$; seed $s$; initial samples $n_{\text{init}}$;
refinement steps $n_{\text{ref}}$

**Ensure:** estimate $\hat{k}$ (or $\varnothing$ if none succeeds)

1: **function** PROBE($k$)
2:     **if** $k \in$ cache **then**
3:         **return** cache[$k$]
4:     **end if**
5:     $(t_k, \text{stats}) \leftarrow$ measure_solve_time$(T, k, \tau, m, s, \dots)$
6:     cache[$k$] $\leftarrow (t_k, \text{stats})$
7:     **return** $(t_k, \text{stats})$
8: **end function**
9: cache $\leftarrow$ empty map
10: **if** $k_{\min} = k_{\max}$ **then**
11:     **return** PROBE($k_{\min}$)
12: **end if**
13: $\mathcal{K}_{\text{sample}} \leftarrow \left\{ \lfloor 10^x \rceil : x \in \text{linspace}(\log_{10} k_{\min}, \log_{10} k_{\max}, n_{\text{init}}) \right\} \cup \{k_{\min}, k_{\max}\}$
14: sort $\mathcal{K}_{\text{sample}}$ and deduplicate
15: $k_{\text{lastOK}} \leftarrow \varnothing, \ k_{\text{upper}} \leftarrow \varnothing$
16: **for all** $k \in \mathcal{K}_{\text{sample}}$ **do**
17:     $(t_k, \_) \leftarrow$ PROBE($k$)
18:     **if** $t_k = \varnothing \ \vee \ t_k > \tau$ **then**
19:         **if** $k_{\text{lastOK}} \neq \varnothing$ **then**
20:             $k_{\text{upper}} \leftarrow k$; **break**
21:         **end if**
22:     **else**
23:         $k_{\text{lastOK}} \leftarrow k$
24:     **end if**
25: **end for**
26: $\hat{k} \leftarrow \arg \min\limits_{(k, t_k) \in \text{cache}, \ t_k \neq \varnothing} \left( |t_k - \tau|, k \right)$
27: **if** $k_{\text{lastOK}} \neq \varnothing$ **and** $k_{\text{upper}} \neq \varnothing$ **then**
28:     $lo \leftarrow k_{\text{lastOK}}, \ hi \leftarrow k_{\text{upper}}$
29:     **for** $i = 1$ **to** $n_{\text{ref}}$ **do**
30:         **if** $hi - lo \leq 1$ **then break**
31:         **end if**
32:         $mid \leftarrow \lfloor (lo + hi)/2 \rfloor$
33:         $(t_{mid}, \_) \leftarrow$ PROBE($mid$)
34:         **if** $t_{mid} \neq \varnothing$ **and** $(|t_{mid} - \tau|, mid) < (|t_{\hat{k}} - \tau|, \hat{k})$ **then**
35:             $\hat{k} \leftarrow mid$
36:         **end if**
37:         **if** $t_{mid} = \varnothing \ \vee \ t_{mid} > \tau$ **then**
38:             $hi \leftarrow mid - 1$
39:         **else**
40:             $lo \leftarrow mid$
41:         **end if**
42:     **end for**
43: **end if**
44: **return** $\hat{k}$

---

 # I   Task Timings

We report the size parameter $n$ and per task timings in Table 9. The average time per task is calculated by running the reference solver on 100 development instances, and repeating this three times.

Table 9: Per task size parameter $n$ values and average time for the reference solve function, across three timing runs.

| Task | n | Average Time (ms) |
| --- | --- | --- |
| affine_transform_2d | 520 | $17.46 \pm 0.14$ |
| articulation_points | 841 | $98.41 \pm 0.48$ |
| base64_encoding | 2387 | $4.41 \pm 0.01$ |
| btsp | 8 | $0.82 \pm 0.02$ |
| capacitated_facility_location | 6 | $263.55 \pm 1.69$ |
| chacha_encryption | 3196 | $0.99 \pm 0.00$ |
| channel_capacity | 138 | $96.18 \pm 0.25$ |
| chebyshev_center | 242 | $101.06 \pm 0.58$ |
| cholesky_factorization | 668 | $8.77 \pm 0.18$ |
| clustering_outliers | 2359 | $92.37 \pm 0.39$ |
| communicability | 61 | $98.25 \pm 1.79$ |
| convex_hull | 191235 | $20.19 \pm 0.33$ |
| convex_quadratic_check | 605 | $100.75 \pm 1.84$ |
| convolve2d_full_fill | 6 | $147.48 \pm 0.23$ |
| convolve_1d | 16325 | $15.32 \pm 0.09$ |
| correlate_1d | 88 | $10.12 \pm 0.04$ |
| count_connected_components | 794 | $16.09 \pm 0.10$ |
| count_riemann_zeta_zeros | 15849 | $68.00 \pm 0.18$ |
| crew_pairing | 343 | $131.36 \pm 1.00$ |
| cumulative_simpson_multid | 67 | $10.01 \pm 0.50$ |
| cyclic_independent_set | 4 | $87.97 \pm 0.17$ |
| determinant_matrix_exponential | 662 | $10.04 \pm 0.02$ |
| dijkstra_from_indices | 1607 | $9.88 \pm 0.02$ |
| discrete_log | 25 | $19.72 \pm 0.16$ |
| dynamic_assortment_planning | 36 | $99.46 \pm 0.94$ |
| earth_movers_distance | 409 | $9.26 \pm 0.05$ |
| edge_expansion | 8507 | $67.99 \pm 0.38$ |
| efficiency | 500 | $101.01 \pm 0.40$ |
| eigenvalues_complex | 476 | $99.02 \pm 0.21$ |
| eigenvalues_real | 358 | $10.10 \pm 0.02$ |
| eigenvectors_complex | 467 | $99.26 \pm 0.08$ |
| eigenvectors_real | 445 | $19.36 \pm 0.13$ |
| elementwise_integration | 378 | $101.44 \pm 0.04$ |
| feedback_controller_design | 12 | $79.83 \pm 0.33$ |
| fft_real_scipy_fftpack | 631 | $8.14 \pm 0.03$ |
| generalized_eigenvalues_complex | 275 | $103.61 \pm 0.05$ |
| generalized_eigenvalues_real | 363 | $20.12 \pm 0.08$ |
| generalized_eigenvectors_complex | 272 | $102.09 \pm 0.21$ |
| generalized_eigenvectors_real | 236 | $9.92 \pm 0.05$ |
| graph_coloring_assign | 42 | $101.39 \pm 0.20$ |
| graph_isomorphism | 127 | $92.04 \pm 0.12$ |
| graph_laplacian | 4839 | $10.03 \pm 0.12$ |
| gzip_compression | 653 | $99.40 \pm 0.01$ |
| job_shop_scheduling | 17 | $99.40 \pm 6.36$ |
| kalman_filter | 23 | $97.41 \pm 0.33$ |
| kcenters | 33 | $9.63 \pm 0.02$ |
| kd_tree | 77 | $10.13 \pm 0.05$ |
| kernel_density_estimation | 631 | $199.79 \pm 0.25$ |
| kmeans | 278 | $96.28 \pm 0.38$ |
| ks_test_2samp | 50610 | $9.97 \pm 0.06$ |
| l0_pruning | 83829 | $9.95 \pm 0.13$ |
| l1_pruning | 96069 | $19.84 \pm 0.22$ |

Continued on next page

| Task | $n$ | Average Time (ms) |
|---|---|---|
| lasso | 87 | $10.14 \pm 0.09$ |
| linear_system_solver | 532 | $10.01 \pm 0.04$ |
| lp_box | 297 | $100.38 \pm 0.59$ |
| lp_centering | 304 | $105.24 \pm 0.32$ |
| lqr | 111 | $99.62 \pm 1.41$ |
| lu_factorization | 497 | $17.62 \pm 0.19$ |
| markowitz | 396 | $103.83 \pm 0.25$ |
| matrix_exponential | 552 | $98.32 \pm 0.68$ |
| matrix_multiplication | 298 | $11.27 \pm 0.09$ |
| matrix_sqrt | 281 | $100.40 \pm 0.21$ |
| max_clique | 17 | $88.86 \pm 2.94$ |
| max_common_subgraph | 4 | $24.21 \pm 0.49$ |
| max_flow_min_cost | 63 | $99.08 \pm 1.20$ |
| max_independent_set | 15 | $55.09 \pm 1.68$ |
| max_weighted_independent_set | 27 | $10.83 \pm 0.82$ |
| min_weight_assignment | 233 | $9.58 \pm 0.01$ |
| minimum_spanning_tree | 574 | $101.43 \pm 0.62$ |
| minimum_volume_ellipsoid | 27 | $115.07 \pm 0.78$ |
| multi_dim_knapsack | 215 | $192.66 \pm 18.89$ |
| nmf | 6 | $83.00 \pm 0.07$ |
| ode_fitzhughnagumo | 17 | $81.52 \pm 0.16$ |
| ode_hires | 706 | $120.77 \pm 0.38$ |
| ode_lorenz96_nonchaotic | 3 | $25.78 \pm 0.23$ |
| ode_nbodyproblem | 9 | $113.80 \pm 0.73$ |
| ode_stiff_robertson | 9999999 | $82.71 \pm 0.15$ |
| ode_stiff_vanderpol | 2 | $111.07 \pm 0.55$ |
| odr | 17637 | $45.08 \pm 0.02$ |
| pagerank | 7978 | $76.79 \pm 0.54$ |
| pca | 36 | $105.81 \pm 0.59$ |
| pde_burgers1d | 12 | $103.34 \pm 0.52$ |
| pde_heat1d | 9 | $98.58 \pm 0.66$ |
| polynomial_mixed | 415 | $103.33 \pm 2.37$ |
| polynomial_real | 396 | $99.12 \pm 0.09$ |
| portfolio_optimization_cvar | 15 | $97.71 \pm 0.24$ |
| procrustes | 307 | $19.76 \pm 0.09$ |
| psd_cone_projection | 349 | $100.60 \pm 0.11$ |
| qp | 278 | $97.59 \pm 0.15$ |
| qr_factorization | 500 | $18.42 \pm 0.56$ |
| quantile_regression | 187 | $98.70 \pm 0.82$ |
| queens_with_obstacles | 14 | $142.63 \pm 3.12$ |
| queuing | 420 | $4.83 \pm 0.01$ |
| qz_factorization | 271 | $99.88 \pm 0.22$ |
| randomized_svd | 310 | $10.23 \pm 0.02$ |
| rbf_interpolation | 85 | $68.17 \pm 0.12$ |
| rectanglepacking | 10 | $341.44 \pm 150.17$ |
| robust_linear_program | 12 | $99.10 \pm 0.41$ |
| rotate_2d | 506 | $18.41 \pm 0.08$ |
| set_cover_conflicts | 9 | $9.90 \pm 1.05$ |
| set_cover | 74 | $154.40 \pm 11.25$ |
| sha256_hashing | 1818 | $1.00 \pm 0.00$ |
| shift_2d | 555 | $18.72 \pm 0.02$ |
| shortest_path_dijkstra | 352 | $100.62 \pm 0.27$ |
| sparse_eigenvectors_complex | 1561 | $183.99 \pm 0.35$ |
| spectral_clustering | 27 | $133.05 \pm 1.21$ |
| stable_matching | 471 | $9.60 \pm 0.03$ |
| toeplitz_solver | 8588 | $101.29 \pm 2.20$ |
| tsp | 39 | $106.21 \pm 26.47$ |
| two_eigenvalues_around_0 | 568 | $19.76 \pm 0.12$ |
| unit_simplex_projection | 127543 | $9.86 \pm 0.10$ |
| vector_quantization | 30 | $9.97 \pm 0.00$ |
| vectorized_newton | 40684 | $4.99 \pm 0.02$ |

| Task | $n$ | Average Time (ms) |
|------|-----|-------------------|
| vehicle_routing | 14 | $91.82 \pm 1.85$ |
| vehicle_routing_circuit | 8 | $140.19 \pm 2.36$ |
| vertex_cover | 1 | $9.83 \pm 1.12$ |
| vertex_cover_cpsat | 17 | $79.95 \pm 1.28$ |
| wasserstein_dist | 15439 | $19.17 \pm 0.11$ |
| water_filling | 4075 | $98.59 \pm 0.65$ |
| zoom_2d | 480 | $26.06 \pm 0.15$ |

1193