# OpenReview forum: "AlgoTune: Can Language Models Speed Up General-Purpose Numerical Programs?"
_NeurIPS.cc/2025/Datasets_and_Benchmarks_Track — NeurIPS 2025 Datasets and Benchmarks Track poster_

### Official Review · Reviewer_RJ6j · 2025-06-25

**Rating:** 5
**Confidence:** 2

**Summary:**

The paper introduces AlgoTune, a novel benchmark designed to evaluate language models' (LMs) capability to optimize numerical functions across various computationally intensive tasks in mathematics, computer science, physics, and machine learning. The benchmark comprises 120 diverse tasks sourced from established Python libraries, supported by comprehensive validation and runtime profiling tools. Additionally, the authors develop AlgoTuner, a baseline LM agent that iteratively optimizes these functions. Although AlgoTuner achieves moderate runtime improvements, it primarily delivers surface-level optimizations rather than novel algorithmic innovations.

**Dataset Code Accessibility:**

Yes

**Ethical Considerations:**

No, there are no or only very minor ethics concerns

**Final Justification:**

The authors have resolved my concern, so I am pleased to raise my rating to accept.

**Limitations Weaknesses:**

1. The evaluation of AlgoTuner is limited by a fixed computational budget, potentially restricting deeper optimization efforts that might uncover more substantial algorithmic improvements.

2. Many tasks, along with their evaluation scripts and verifiers, appear to have been created or refined manually. This raises concerns about human bias, potential overfitting, and lack of scalability. It would strengthen the work to introduce automatic task generation or task-agnostic validation mechanisms that scale to new problems.

3. Also, the reliance on a predetermined set of test inputs for solution verification presents a potential vulnerability. If the input distribution is insufficiently comprehensive or the verifier lacks completeness, there is a risk of incorrect yet seemingly performant LM-synthesized code passing validation. Future research may explore more stringent verification methodologies, acknowledging their inherent complexities.

4. While the paper positions AlgoTuner as a baseline, its contributions appear incremental and mostly at the systems level. It relies heavily on prompting and does not seem to explore deeper innovations such as learning to adapt search strategies or modeling optimization trajectories. I would encourage the authors to compare with RL-based code optimization agents or more advanced planning methods.

**Strengths Contributions:**

1. Novel Benchmark: AlgoTune is the first benchmark I am aware of that explicitly evaluates both correctness and runtime performance in code generated by LMs for numerical optimization problems. This dual emphasis is essential for deploying LMs in real-world scientific and engineering workflows.

2. Diverse Task Set: The authors have meticulously curated 120 tasks spanning multiple disciplines, which are much more representative of real-world needs than toy problems. The use of well-established libraries like SciPy and NetworkX lends credibility and reproducibility.

3. Robust Evaluation Framework: The runtime profiling setup, including time and memory measurements, adds rigor. The authors’ framework accounts for warm-up effects and avoids common measurement pitfalls. This enables more precise and trustworthy evaluation of LM-generated code beyond correctness.

---

> ### Author Rebuttal · Authors · 2025-07-30
>
> We thank you for your feedback, and are pleased to see you found our benchmark novel, the tasks diverse, and the evaluation robust. We now address the points you brought up:
>
> >The evaluation of AlgoTuner is limited by a fixed computational budget, potentially restricting deeper optimization efforts that might uncover more substantial algorithmic improvements.
>
> The fixed budget is intended to provide an apples to apples comparison between the models. We re-ran all four language models using AlgoTuner on all tasks, using double the budget (＄1 instead of ＄0.50) and only noticed a mild increase in AlgoTune score (for example, AlgoTuner with o4-mini went from 1.58x to 1.71x). While the models come up with some strong optimization (showcased in Section 4.2), they still do not manage to find substantially novel algorithms. This shows that there's much room for improvement on building better systems for AlgoTune, and we're excited to see how our benchmark can measure and guide development in this research direction. We have updated the paper with these results.
>
> >Many tasks, along with their evaluation scripts and verifiers, appear to have been created or refined manually [...]
>
> AlgoTune’s tasks are manually curated by human experts in order to collect tasks that are interesting and well defined. During the writing process we experimented in automatic task generation, but this almost always produced tasks that were uninteresting or not well defined, and the automatically produced tests were typically of low quality to the point of being unusable.
> Regarding overfitting: This is a valid concern and we took precautions in our experiment design. We show the following experiment, omitted in the original paper due to space constraints: AlgoTuner has no direct access to dev or test instances; to determine if there is overfitting on the dev samples, we show the difference between dev and test performance to be negligible in the following table:
>
> | Model Name     | Median Dev/Test Offset |
> |----------------|------------------------|
> | R1             | +0.016x                |
> | o4-mini        | +0.004x                |
> | Gemini 2.5 Pro | -0.021x                |
> | Claude Opus 4  | +0.000x                |
>
> **Note:**
> The offset is defined as *(Dev Speedup / Test Speedup − 1)*, computed per task.
> Values shown are the median offset across tasks for each model.
> Positive = Dev performed better than Test; Negative = Test outperformed Dev.
>
>
> >If the input distribution is insufficiently comprehensive or the verifier lacks completeness, there is a risk of incorrect yet seemingly performant LM-synthesized code passing validation [...]
>
> We agree that code optimization depends on the function being optimized as well as the input data distribution. For this reason, we used human experts to define logical data distributions for each task. Each task was accompanied by tests, which were then given to 2 reviewers to red-team to make sure the coverage was wide. We agree with the reviewer that future work could explore ways to fully execute or enhance this approach with agents.
>
> >While the paper positions AlgoTuner as a baseline, its contributions appear incremental [...]
>
> To align with the Datasets and Benchmarks track, our core contribution is the AlgoTune benchmark, and not the AlgoTuner baseline. We agree that our AlgoTuner baseline is simpler than other code optimization systems one could build, and we do that on purpose, to show what an initial baseline model could achieve. This is similar to other code benchmarking papers including SWE-bench and Humanity's Last Exam, that provided only simple baselines in their initial papers and did not develop deep technical systems with their initial benchmarks. We have changed the wording in the introduction and abstract to make this more clear.

---

### Official Review · Reviewer_JKfv · 2025-07-01

**Rating:** 4
**Confidence:** 3

**Summary:**

AlgoTune is a new benchmark designed to evaluate language models' ability to design and implement efficient algorithms for challenging problems in CS, physics, and math, beyond tasks already solved by humans. It includes 120 expert-designed tasks and a testing framework. A baseline agent, AlgoTuner, achieves a 1.58× speedup over standard solvers, but current models mostly perform shallow optimizations rather than discovering novel algorithms. The benchmark aims to spur the development of more creatively capable LMs.

**Dataset Code Accessibility:**

Yes

**Dataset Code Comments:**

Yes, I have checked the code.

**Ethical Considerations:**

No, there are no or only very minor ethics concerns

**Final Justification:**

The response address most of my questions, so I keep my recommendation of 4.

**Limitations Weaknesses:**

- Section 2 is very long, and it is suggested to give a brief outline at the beginning or have more subsections.
- What is the optimality test mentioned in l162?/？
- Do you think that the budget limit is one of the main reasons why the current agent can not discover algorithmic innovations? Also, from Figure 3, the scores are still increasing with the budget.

**Strengths Contributions:**

- The paper proposes a new task for speeding up numerical algorithms with LMs, which is novel and important.
- The authors collect many numerical algorithms and normalize them.
- The paper also designs the AlgoTuner Agent for this problem and achieves success.

---

> ### Author Rebuttal · Authors · 2025-07-30
>
> We thank you for your feedback, and were happy to see that you found AlgoTune novel and important, and AlgoTuner useful.
>
> >Section 2 is very long, and it is suggested to give a brief outline at the beginning or have more subsections.
>
> Thank you for mentioning this: we've added an outline at the beginning of the section and have separated the content in Section 2 into 3 subsections: Benchmark Scope, Task Implementation, and Evaluation Protocol.
>
> Each subsection has been given proper titles, and the revised paper now has the following outline at the start of Section 2:
>
>
> This section defines the benchmark scope (domains, taxonomy, and design principles) (§2.1), explains task construction (generators, reference solvers, and verifiers) (§2.2), and details the evaluation protocol (instance sizing, timing, and metrics) (§2.3). Together, these subsections specify how tasks are generated, how correctness and speedups are measured, and how results can be reproduced.
>
> >What is the optimality test mentioned in l162?/？
>
>
> As part of the is_solution function, the optimality test for vector quantization checks not only that the solution found is valid, but that it is also an optimal solution.
>
>
> >Do you think that the budget limit is one of the main reasons why the current agent can not discover algorithmic innovations?  [...]
>
>
> Thank you for this suggestion. We re-ran all four language models using AlgoTuner on all tasks, using double the budget (＄1 instead of＄0.50) and only noticed a mild increase (from 1.58x to 1.71x). While the models come up with some strong optimizations (showcased in Section 4.2), like in previous experiments, they do not manage to find substantially novel algorithms. This shows that there's much room for improvement on AlgoTune, and we're excited to see how AlgoTune can measure and guide development in this research direction. We have updated the revised paper with these results.

---

### Official Review · Reviewer_hjPK · 2025-07-02

**Rating:** 5
**Confidence:** 3

**Summary:**

This paper introduces AlgoTune, a benchmark designed to evaluate the ability of large language models (LMs) to optimize general-purpose numerical programs. The benchmark consists of 120 challenging computational problems sourced from experts in computer science, physics, and mathematics. The authors also developed "AlgoTuner," an LM-based agent that iteratively writes and refines code to solve these problems efficiently. The core contribution is the evaluation of current frontier LMs on their ability to go beyond known solutions and discover algorithmic improvements for computationally intensive tasks. The authors find that while their AlgoTuner agent achieves a notable 1.58x average speedup over established open-source libraries, the optimizations are primarily "surface-level" rather than true algorithmic innovations.

**Dataset Code Accessibility:**

Yes

**Dataset Code Comments:**

Code is provided.

**Ethical Considerations:**

No, there are no or only very minor ethics concerns

**Final Justification:**

The authors have addressed all of my concerns. I have increased my score to accept. The following issues are resolved:
comparison with mercury, failure analysis, generalization issues, qualitative analysis of the proposed agent, with apt explanations.

**Limitations Weaknesses:**

* In abstract, it would be good to add more information about how AlgoTuner is developed: “In addition, we develop a baseline LM agent, AlgoTuner, and evaluate its performance across a suite of frontier models.”.
* Comparison with Mercury (Neurips 2024 benchmarking and datasets), which assesses both correctness and efficiency of the generated code, one can identify correlation with mercury benchmark to identify whether there is any new signal that algotune is providing.
* It would be great to include error bars in Figure 3, as the gaps are small across llms. Code generation/refinement is inherently stochastic.
* The AlgoTune benchmark evaluates each task at a single problem size (n), which is a critical limitation because the optimal algorithm often depends on the input scale . The authors acknowledge this on page 5, lines 180-182 .By fixing n to a value that results in a 100ms runtime for the reference solver, the benchmark may reward optimizations that are only effective at that specific scale, rather than measuring true asymptotic efficiency . This means the evaluation measures performance at a single, arbitrary point, which limits the generalizability of the reported speedups. An agent could achieve a speedup on a small n with a solution that is asymptotically worse and would be significantly slower on larger problem sizes.
* The "Failure Analysis" section is brief and anecdotal, presenting only two specific examples: the sinkhorn task and the portfolio_optimization_cvar task. For a benchmark paper of this scope, a more systematic and quantitative breakdown of failure modes is essential. The paper leaves many critical questions unanswered. For instance, across the 120 tasks, what percentage resulted in no speedup at all? How many generated code that was slower than the reference? How many produced incorrect solutions that were caught by the verifier? In how many cases did the agent simply exhaust its budget without finding a valid improvement? A more comprehensive analysis, perhaps correlating failure types with task categories or the complexity of the reference implementation, would provide a much clearer and more valuable picture of the current limitations of LM-based optimization.
* The paper includes a plot (Figure 3) showing performance scaling with the monetary budget, which is a useful inclusion. However, this analysis is performed on the development set, not the held-out test set. This makes it impossible to check for potential overfitting, where the agent might be tuning its solution specifically to the development examples.
* The analysis of  what the agent learns during its iterative process is absent. Does it try simple, high-probability optimizations first (like adding a Numba JIT decorator)? Does it learn from its mistakes, such as generating incorrect code and then correcting it? A deeper analysis of the agent's optimization trajectory would provide valuable insights into the reasoning and problem-solving processes of these systems.
* The central finding that current systems fail at finding new algorithmic innovations. This could be because of the search strategy used by the agent. AlgoTuner agent uses an iterative edit-and-verify loop, which is a form of local search. This method is effective for finding surface-level optimizations close to the starting solution. However, true algorithmic innovation, like switching from a simple to a complex sorting algorithm, requires a large, non-local leap in the program space. A local search strategy is fundamentally ill-equipped for such jumps. In contrast, systems like DeepMind's FunSearch use an evolutionary algorithm—a form of global search—that maintains a diverse population of programs and uses the LLM to creatively mutate them. This approach is designed to explore different parts of the solution space and can lead to algorithmic discoveries


Questions to authors:
Is there a reason why the number 120 was chosen, any limitation in scaling it up to 1000 examples?

**Strengths Contributions:**

* The paper addresses a critical and forward-looking question: can LMs contribute to algorithmic discovery and optimization in a meaningful way? This moves beyond the typical evaluation of LMs on tasks with existing, known solutions and pushes the research frontier toward creative problem-solving.

* Curating 120 problems from domain experts provides a diverse and challenging testbed for future research in this area. The inclusion of a framework for validation and timing is also a valuable practical contribution.

* The finding that the "AlgoTuner" agent can achieve a 1.58x speedup over highly optimized, popular libraries like SciPy and scikit-learn is impressive and demonstrates the practical potential of this approach, even in its early stages.

---

> ### Author Rebuttal · Authors · 2025-07-30
>
> We thank you for your feedback. We were pleased to see that you found that AlgoTune addresses a critical question for language model evaluation, and our tasks to be diverse and challenging. We address your concerns below:
>
> >In abstract, it would be good to add more information about how AlgoTuner is developed
>
> Thanks for your suggestion. We have added the following line to the abstract:
>
> AlgoTuner uses a simple, budgeted loop that edits code, compiles and runs it, profiles performance,verifies correctness on tests, and selects the fastest valid version.
>
> >Comparison with Mercury (Neurips 2024 benchmarking and datasets) [...]
>
> Thank you for bringing this relevant paper to our attention. Whereas Mercury includes LeetCode-style coding problems, i.e. quicksort, two-sum, and so on, to the best of our knowledge we are the first benchmark to focus on practical algorithms with broad, real-world deployment to have practical relevance for mainstream software engineering and computational sciences. AlgoTune includes programs including AES encryption, gzip compression, SVD matrix decomposition, and more.
>
> Thank you again for this reference. We apologize for the omission and have corrected the revised version of our paper with a citation and discussion of Mercury and its relationship to our work.
>
> >It would be great to include error bars in Figure 3 [..]
>
> Thank you for this great suggestion, we’ve added it to the revised paper.
>
> >The AlgoTune benchmark evaluates each task at a single problem size [..]
>
> We agree that the optimal solution for a given problem depends on the size of the problem instance. AlgoTune is the first benchmark that is built with this in mind, and allows for generation of problem instances in arbitrary sizes. In the interest of reducing profiling noise and standardizing costs across tasks and models, we fix per‑task problem sizes such that the reference solver runs in 100 ms. Adding variable problem instance sizes would increase the complexity, money, and time required to run the current benchmark and so in this initial exploration we decided to not pursue this direction. We hope that this initial design motivates and enables follow‑up work that explicitly studies optimization tradeoffs across input scales, something which we agree is unfortunately absent in the literature today.
>
> >The "Failure Analysis" section is brief and anecdotal [...]
>
> Thanks for this great suggestion. We have added the following analysis tables to the revised paper:
>
> We analyzed the distribution of solution outcomes:
>
> | Model | **Speedup (share of tasks)** | | | |
> |:--|:--:|:--:|:--:|:--:|
> | | ≥1.1× | 0.9×–1.1× | <0.9× | Invalid |
> | Claude Opus 4 | 40.0% | 32.9% | 2.6% | 24.5% |
> | DeepSeek R1 | 61.3% | 25.7% | 6.5% | 6.5% |
> | o4-mini | 60.0% | 32.3% | 4.5% | 3.2% |
> | Gemini 2.5 Pro | 49.0% | 17.4% | 6.5% | 27.1% |
> | **Overall (Average)** | **52.6%** | **27.1%** | **5.0%** | **15.3%** |
> | **Overall (Task Best)** | **71.7%** | **23.2%** | **3.2%** | **1.9%** |
>
> Note:
> - **Overall (Average)**: Performance across all individual attempts
> - **Overall (Task Best)**: Performance if we could pick the best model for each task
> - Invalid: code produced invalid outputs or timed out on at least 1 instance
>
> In summary:
> In 15.3% cases, the models exhausted their budget without finding any valid solution. In 71.7% of tasks, a speedup of at least 1.1x was found by at least one model.
>
> >The paper includes a plot (Figure 3) showing performance scaling with the monetary budget, which is a useful inclusion [...] This makes it impossible to check for potential overfitting [...]
>
> Thank you for this, we ran the following experiment to check overfitting:
> We note: AlgoTuner does not have direct access to the train or test samples. To measure overfitting, we look at the median ratio of speedups at the end of each run, on both the dev and the test instances for each model. In the following table, we show this difference to be negligible. This leads us to conclude that there is no overfitting occuring:
>
> | Model Name     | Median Dev/Test Offset |
> |----------------|------------------------|
> | R1             | +0.016x                |
> | o4-mini        | +0.004x                |
> | Gemini 2.5 Pro | -0.021x                |
> | Claude Opus 4  | +0.000x                |
>
> **Note:**
> The offset is defined as *(Dev Speedup / Test Speedup − 1)*, computed per task.
> Values shown are the median offset across tasks for each model.
> Positive = Dev performed better than Test; Negative = Test outperformed Dev
>
> >The analysis of what the agent learns during its iterative process is absent [...]
>
> Thanks for your suggestion. We have added the following tables to the revised paper:
>
> In this table, we look at how the speedups found by AlgoTuner changed during the iterations. The following shows a per-model breakdown of the trajectories, based on how the first attempted code that ran without errors went, versus the best code.
>
> - Significant: Speedup ≥1.1x
> - Insignificant: 0.9x <= Speedup < 1.1x
> - Slow: Speedup < 0.9
>
> | Model | Always Significant | Invalid to Significant | Insignificant to Significant | Slow to Significant | Always Insignificant | N/A to Insignificant | Always N/A |
> |-------|----------------------------|---------------------|---------------------------|------------------|---------------------|----------------------|-------------------|
> | Claude Opus 4 | 16.1% | 7.7% | 16.8% | 1.9% | 31.6% | 5.2% | 20.7% |
> | DeepSeek R1 | 27.7% | 19.4% | 14.8% | 7.1% | 11.0% | 11.0% | 9.0% |
> | Gemini 2.5 Pro | 36.1% | 9.7% | 9.7% | 2.5% | 12.3% | 4.5% | 25.2% |
> | o4-mini | 34.8% | 16.1% | 7.8% | 4.5% | 14.8% | 14.8% | 7.2% |
> | **Overall** | **28.7%** | **13.2%** | **12.3%** | **4.0%** | **17.4%** | **8.9%** | **15.5%** |
>
>
> Lastly, restricting ourselves to tasks that were sped up by at least 1.1x, we look at how much of the budget was exhausted to get to 1) the initial speedup, and 2), the maximum speedup.
>
> | Model | $ to First Speedup | $ to Max Speedup | # of New Max Speedups | Median Offset Ratio (Max/First - 1) |
> |---|---|---|---|---|
> | Claude Opus 4 | $0.38 | $0.50 | 0 | 0.00x |
> | DeepSeek R1 | $0.05 | $0.59 | 3 | 0.22 |
> | Gemini 2.5 Pro | $0.13 | $0.46 | 1 | 0.04 |
> | o4-mini | $0.08 | $0.46 | 2 | 0.30 |
> | **Overall** | **$0.12** | **$0.51** | **1** | **0.09** |
>
> All reported numbers are medians across all tasks for each model. # of New Max Speedups refers to the median number of max speedups after the initial speedup of at least 1.1x.
> We note that these results are done on AlgoTuner runs with double the budget used in the submitted version (from ＄0.50 to ＄1), which have been added to the revised paper.
>
> >Does it try simple, high-probability optimizations first (like adding a Numba JIT decorator)? Does it learn from its mistakes, such as generating incorrect code and then correcting it?
>
> Thanks for this suggestion. To get more insights on the solutions proposed by AlgoTuner, we extend Table 3. We have added the following table (for all language models tested) to the revision.
>
> Here, we look at the packages used by AlgoTuner using o4-mini, and how they change from the first solution that gets a speedup >= 1.1x, to the solution that gets the maximum speedup. We also note the packages used in the reference solution. Interestingly, JAX’s jit is used quite a lot in the reference solutions, but not so much in the first speedup. The model then uses it a lot in the max speedup solution. Apart from adding these packages, we note that none of the models removed any packages used in the initial solution, i.e., there were no tasks with negative deltas.
>
> | Package | Reference | First ≥1.1x | Max Speedup | Δ (Max-First) |
> |---------|-----------|-------------|-------------|-------------|
> | jax_jit | 155 | 96 | 155 | +59 |
> | numpy | 132 | 73 | 121 | +48 |
> | scipy | 61 | 39 | 73 | +34 |
> | cvxpy | 27 | 2 | 9 | +7 |
> | sklearn | 9 | 3 | 5 | +2 |
>
> >The central finding that current systems fail at finding new algorithmic innovations [...] However, true algorithmic innovation, like switching from a simple to a complex sorting algorithm, requires a large, non-local leap in the program space. A local search strategy is fundamentally ill-equipped for such jumps. [...]
>
> To align with the Datasets and Benchmarks track, our core contribution is the AlgoTune benchmark. We agree that our AlgoTuner baseline is simpler than other code optimization systems one could build, and we do that on purpose, to show what an initial baseline model could achieve. This is similar to other code benchmarking papers including SWE-bench and Humanity's Last Exam, that provided only simple baselines in their initial papers and did not develop deep technical systems with their initial benchmarks. We will change the wording in the intro and abstract to make this more clear.
> Just like SWE-bench and Humanity's Last Exam motivated the community to build novel technical solutions to those tasks, we hope that Algotune will inspire the community to work towards AI systems that “think outside the box” and come up with novel algorithms. And we hypothesize that the community will explore many avenues to get there, including of course the evolutionary search approach that you mentioned.
>
> >Is there a reason why the number 120 was chosen [..]
>
> AlgoTune’s tasks are manually curated by human experts in order to collect tasks that are interesting and well defined. On average, adding each task took around 5 hours, which includes developing task-specific stringent tests and realistic data generation pipelines. Each task was written by an expert and reviewed by at least two others. Scaling this up to 1000 examples would be beyond the time and monetary budget for an academic paper.
> During the writing process we experimented in automatic task generation, but this almost always produced tasks that were uninteresting or not well defined, and the automatically produced tests were typically of low quality to the point of being unusable.

---

### Official Review · Reviewer_uKUs · 2025-07-03

**Rating:** 4
**Confidence:** 3

**Summary:**

This paper proposes testing language models’ ability to design and implement algorithms through an open-ended benchmark called AlgoTune. Specifically, the authors task language models (LMs) with writing efficient code to solve computationally challenging problems from computer science, physics, and mathematics. The AlgoTune benchmark consists of 120 coding tasks gathered from domain experts. Additionally, the authors introduce a baseline LM agent, AlgoTuner, and evaluate its performance using various advanced LMs. AlgoTuner demonstrates an average 1.58x speedup compared to reference solvers.

**Dataset Code Accessibility:**

Yes

**Ethical Considerations:**

No, there are no or only very minor ethics concerns

**Final Justification:**

I have read the response and the other reviews. I will keep my original score.

**Limitations Weaknesses:**

1.	The evaluation tasks presented in the paper share substantial similarities with recent, prominent works in the scientific discovery agent domain, such as FunSearch and AlphaEvolve. However, the paper lacks a detailed and explicit discussion highlighting the relationship, distinctions, and specific contributions of AlgoTune compared to these existing benchmarks and methodologies.
2.	The novelty of the proposed benchmark appears limited, as it primarily comprises manually collected coding tasks (120 tasks) and utilizes an existing SWE-Agent-based method. Consequently, the incremental nature of the contribution somewhat constrains the overall impact and originality of the work.

**Strengths Contributions:**

1.	The proposed benchmark evaluating LLMs’ capability to efficiently write code for challenging computational problems spanning multiple scientific domains is intriguing and addresses an important research direction.
2.	The AlgoTune benchmark provides a well-constructed dataset of 120 expert-curated coding tasks. Furthermore, the authors present AlgoTuner, an LM-based baseline agent that achieves notable performance improvements, specifically an average 1.58x speedup over existing reference solvers, demonstrating the practical potential of this approach.

---

> ### Author Rebuttal · Authors · 2025-07-30
>
> We thank you for your review. We’re happy you found AlgoTune to be intriguing and important. We address your comments below:
>
> > paper lacks a detailed and explicit discussion highlighting the relationship, distinctions, and specific contributions of AlgoTune compared to these existing benchmarks [FunSearch, AlphaEvolve]
>
> Thank you for your suggestion. FunSearch and AlphaEvolve both show how AI systems can improve the quality of solutions on a limited (4 and 13, respectively) set of published problems spanning just a handful of mathematical domains. AlgoTune contains 120 problems spanning 12+ categories (Table 1) in Math/Physics/Computer Science. Unlike these 3 previous benchmarks which challenge LMs to optimize solution quality (i.e. finding a shorter TSP path), AlgoTune challenges LMs to implement a given algorithm faster (i.e. write the fastest program to find a TSP path). This allows us to benchmark widely used functions where the optimal solution is already known (like finding eigenvectors, encryption), but the optimal algorithm to get to that solution isn’t.
>
> We’ve cited both FunSearch and AlphaEvolve and clarified the difference to our work in Section 5 in the revised paper.
>
>
> > The novelty of the proposed benchmark appears limited [..]
>
> AlgoTune’s novelty comes from its diversity of tasks, spanning 12+ domains, with problems coming from Math, Physics, and Computer Science. AlgoTune is the first benchmark that consists of widely used functions from a variety of popular Python repositories, making improvements on AlgoTune easily translatable into real-world speedups. Whereas other code optimization benchmarks feature on mathematical problems for which the optimal solution is unknown (FunSearch, AlphaEvolve), or LeetCode style problems (HumanEval, Mercury), AlgoTune features gzip compression, AES encryption, and PageRank, among others. These functions are essential to internet traffic and are used by virtually every application that connects to the internet. Speeding them up is not trivial, and would have instant, wide-reaching benefits.

---

> > ### Comment · Reviewer_uKUs · 2025-08-09
> >
> > Thanks for the detailed response. I will maintain my original score.

---

### Author Response · Authors · 2025-08-04

Dear Reviewers,

As the discussion period nears its end, we wanted to reach out to see if you still had unanswered concerns. We would be happy to use this time to resolve any final issues you may have.

Thanks,

The Authors

---

### Note · Authors · 2025-08-12

Dear AC,

Reviewers found AlgoTune important (JKfv, uKUs), valuable (hjPK), and novel (RJ6j).

Two minor issues were raised, which we addressed in our response:
1. Reviewers questioned if the AlgoTuner budget was too low: We therefore increased the per-task budget from ＄0.5 to ＄1 and re-ran all the experiments. This led to only slightly improved results (1.58x → 1.71x), indicating that AlgoTune is not trivially solved by more compute.

2. Reviewers asked if AlgoTuner-generated code potentially overfit to the dev input set: We analyzed dev vs test set performance empirically and showed that there are no signs of overfitting


No further questions were raised after our rebuttal.

Our main contribution is the AlgoTune benchmark, challenging LMs to optimize the speed of widely used functions from computer science, math, and physics.

Our baseline agent, AlgoTuner, is deliberately simple, as in the baselines released in other benchmarks such as SWE-Bench and Humanity’s Last Exam. This leads to the AlgoTune benchmark cleanly evaluating the foundational models themselves, and not the human-built scaffolding around them.

Achieving significant gains on AlgoTune will require models that will be capable of devising efficient, novel algorithms, and such improvements will lead to real-world impact, as the functions in our benchmark are widely-used in practice. We are excited to see the community develop language models that are able to further optimize these already highly-efficient functions.

Thanks!

---

### Decision · Program_Chairs · 2025-09-18

**Decision:**

Accept (poster)

**Comment:**

The paper presents a benchmark of tasks for language models to solve with code (focus on numerical programs), along with a baseline agent for solving the tasks. The reviewers praise the wide variety of tasks and the quality of the baseline algorithm provided. While some complaints about novelty are given, the authors point out in their rebuttal that related works do not offer nearly as many coding tasks as the AlgoTune benchmark does. The topic is highly relevant to the NeurIPS community and the benchmark is of high quality. It ought to be accepted.